# HiBaNG: Hierarchical Bayesian Nonparametric Granger Causal Discovery in Low-Data Regimes

**He Zhao**                                    *he.zhao@data61.csiro.au*
*CSIRO's Data61, Australia*
*& Department of Data Science and AI, Monash University, Australia*

**Vassili Kitsios**                            *vassili.kitsios@csiro.au*
*CSIRO's Environment, Australia*
*& Laboratory for Turbulence Research in Aerospace and Combustion,*
*Department of Mechanical and Aerospace Engineering, Monash University, Australia*

**Terence J. O'Kane**                          *terence.okane@csiro.au*
*CSIRO's Environment, Australia*
*& University of Tasmania, Australia*

**Edwin V. Bonilla**                           *edwin.bonilla@csiro.au*
*CSIRO's Data61, Australia*

**Reviewed on OpenReview:** *https://openreview.net/forum?id=e4VO3YlRBr*

## Abstract

We present a principled probabilistic framework for discovering Granger causal relationships from multivariate time-series data in low-data regimes, where short sequences limit the applicability of modern deep learning approaches. While deep neural vector autoregressive (VAR) models perform well in high-data settings, they often struggle to generalize with limited samples and provide little insight into model uncertainty. To address these challenges, we introduce HiBaNG, a hierarchical Bayesian nonparametric framework for Granger causal discovery. HiBaNG places a hierarchical factorized prior over binary Granger causal graphs that encodes structured sparsity and enables interpretable, uncertainty-aware inference. We develop a tractable Gibbs sampling algorithm that exploits conjugacy and augmentation for scalable posterior estimation. Extensive experiments on synthetic, semi-synthetic, and real-world climate datasets demonstrate that HiBaNG consistently outperforms both classical and deep VAR baselines, achieving improved accuracy and calibrated uncertainty.

## 1 Introduction

Multivariate time-series (MTS) data consist of observations of multiple variables recorded at multiple timestamps and are fundamental to a wide range of applications in economics, healthcare, climatology, and neuroscience. In these domains, uncovering causal relationships among time-series is often essential for understanding system dynamics and supporting decision-making.

In this paper, we focus on discovering such relationships from observational MTS data using Granger Causality (GC) (Granger, 1969; Lütkepohl, 2005; Shojaie & Fox, 2022), which posits that one variable is causal for another if its past values provide statistically significant information for predicting the future of the latter, beyond what is contained in its own past. While GC does not necessarily imply a structural or interventional causal relationship in the sense of, e.g., Pearl's do-calculus (Pearl, 2009) or Rubin's potential outcomes (Imbens & Rubin, 2015), it remains widely used for time-series causality[1]. The most common implementation of GC

---

[1]Throughout this paper, causality refers specifically to Granger causality.

is via Vector Autoregressive (VAR) models (Lütkepohl, 2005), which assume that each variable is a function of the lagged values of other variables.

Bayesian VARs (Woźniak, 2016; Miranda-Agrippino & Ricco, 2019) extend classical VARs by incorporating prior beliefs, enabling uncertainty quantification and improved estimation under limited data. More recently, deep learning-based VAR models (Montalto et al., 2015; Tank et al., 2018; Wang et al., 2018; Nauta et al., 2019; Khanna & Tan, 2020; Wu et al., 2020; Marcinkevičs & Vogt, 2021; Gong et al., 2022; Fan et al., 2023) have gained popularity for their ability to capture complex, nonlinear dynamics, provided ample data are available. In practice, however, many real-world applications do not have the luxury of abundant data, especially during the early stages of data collection. In these low-data regimes, where the number of samples is small relative to the number of variables, deep VARs often underperform. This is mainly due to the following: **1)** When model complexity exceeds the informational content of the data, the learned function becomes unstable, noise-sensitive, and yields unreliable predictions with distorted uncertainty estimates. (Geffner et al., 2022; Annadani et al., 2023; Deleu et al., 2023); **2)** In practice, model selection and hyperparameter tuning are difficult in causal discovery because the task is unsupervised: the target object (the true causal graph) is typically unobserved, and therefore cannot be used directly as a validation target Biza et al. (2020). Moreover, although ground-truth graphs may exist in synthetic benchmarks, using them for model tuning can lead to potentially biased comparisons, since it amounts to selecting models with privileged access to evaluation information that is unavailable in real applications Machlanski et al. (2023).

To address these challenges, we propose a new hierarchical Bayesian VAR framework for Granger causal discovery, specifically designed for low-data regimes. Our method addresses the aforementioned issues by:

1. Introducing a hierarchical factorized prior over binary Granger causal graphs, which is designed to encode structured sparsity, adaptive regularization, and low-complexity latent organization, helping stabilize causal discovery in low-data regimes where likelihood-based evidence alone is insufficient;

2. Decomposing Granger causality into discrete (graph structure) and continuous (causal strength) components, allowing the binary graph to constrain parameter estimation and improve generalization;

3. Leveraging Bayesian nonparametric techniques to integrate out latent factors and reduce the number of tunable hyperparameters.

Overall contribution: We develop **HiBaNG**, a **hi**erarchical **Ba**yesian **n**onparametric framework for **G**ranger causal discovery in data-scarce settings, which integrates interpretable priors, principled uncertainty quantification, and enables tractable posterior inference via Gibbs sampling. Our extensive experiments on synthetic, semi-synthetic, and real-world climate datasets show that HiBaNG attains improved or competitive performance relative to both classical and deep VAR baselines.

## 2 Preliminaries

Consider a collection of MTS data consisting of $N$ time series or variables and $T$ timestamps, stored in the matrix $\boldsymbol{X} \in \mathbb{R}^{N \times T} = (\boldsymbol{x}_1, \ldots, \boldsymbol{x}_T)$, where $\boldsymbol{x}_t \in \mathbb{R}^N$ consists of the samples/values of the $N$ variables at timestamp $t \in \{1, \ldots, T\}$. A VAR model for Granger causality (Lütkepohl, 2005; Hyvärinen et al., 2010) assumes that $\boldsymbol{x}_t$ can be predicted from the $\tau_{\max}$ time lags $\{\boldsymbol{x}_{t-1}, \ldots, \boldsymbol{x}_{t-\tau_{\max}}\}$ by learning a coefficient matrix $\boldsymbol{A}^\tau \in \mathbb{R}^{N \times N}$ for each lag $\tau \in \{1, \ldots, \tau_{\max}\}$:

$$\boldsymbol{x}_t = \sum_{\tau=1}^{\tau_{\max}} \boldsymbol{A}^{\tau 2} \boldsymbol{x}_{t-\tau} + \epsilon_t, \tag{1}$$

where $\epsilon_t$ is an independent noise term. Conventionally, variable $j$ (the parent) does not Granger-cause variable $i$ (the child) ($i, j \in \{1, \ldots, N\}$) if and only if for all $\tau$, $A_{ij}^\tau = 0$. For deterministic VARs, learning can be done by minimizing a regression error: $\min_{\{\boldsymbol{A}^\tau\}_\tau^{\tau_{\max}}} \|\boldsymbol{x}_t - \sum_{\tau=1}^{\tau_{\max}} \boldsymbol{A}^\tau \boldsymbol{x}_{t-\tau}\|_2^2 + \lambda \operatorname{reg}(\{\boldsymbol{A}^\tau\}_\tau^{\tau_{\max}})$, where $\operatorname{reg}(\{\boldsymbol{A}^\tau\}_\tau^{\tau_{\max}})$ is a sparsity-inducing penalty (e.g., a group lasso penalty (Yuan & Lin, 2006; Lozano et al., 2009)): $\sum_{i=1,j=1}^N \|A_{i,j}^\tau\|_2$. Other alternative penalties can be found in Nicholson et al. (2017).

---

[2]We use superscripts to index matrices, rather than to denote powers.

**Bayesian VARs** (BVARs) (Litterman, 1986) are another important line of research, especially in econometrics and finance. A standard method may model $\boldsymbol{x}_t$ with a multivariate normal distribution: $\boldsymbol{x}_t \sim \mathcal{MN}\left(\sum_{\tau=1}^{\tau_{\max}} \boldsymbol{A}^\tau \boldsymbol{x}_{t-\tau}, \Sigma\right)$, where various priors can be imposed on $\{\boldsymbol{A}^\tau\}_{\tau=1}^{\tau_{\max}}$ (e.g., sparsity-inducing priors) and $\Sigma$ (e.g., inverse-Wishart priors). Learning BVARs involves inferring the posterior of $\{\boldsymbol{A}^\tau\}_{\tau=1}^{\tau_{\max}}$ and $\Sigma$. In standard BVARs, one may need to "convert" $\{\boldsymbol{A}^\tau\}_{\tau=1}^{\tau_{\max}}$ into GC graphs.

**Deep VARs** have recently become popular, as they use deep neural networks to model nonlinear dynamics across timestamps, which essentially generalize Eq. (1) using: $x_{it} = f_i\left(\sum_{\tau=1}^{\tau_{\max}} \boldsymbol{A}^\tau \boldsymbol{x}_{t-\tau}\right) + \epsilon_t$, where $f_i$ is typically implemented as a nonlinear neural network.

## 3 Method

In this section, we first give a complete picture of our proposed model and then discuss the motivations for our design choices at the end of the section. Specifically, we first present our BVAR model in a general form, which separates the coefficients into binary GC graphs and weight matrices; then we propose a new link function that helps build a hierarchical model on binary GC graphs in Section 3.1; subsequently, we provide details of the full model in Section 3.2 and its Bayesian inference algorithm in Section 3.3. Finally, in Section 3.4, we discuss some properties and design choices in our model that allow us to tackle the challenges mentioned in the introduction.

Our overall model can be seen as a Bayesian VAR of the form:

$$\boldsymbol{x}_t \sim \mathcal{MN}\left(\sum_{\tau=1}^{\tau_{\max}} \left(\boldsymbol{A}^\tau \odot \boldsymbol{G}^\tau\right) \boldsymbol{x}_{t-\tau}, \Sigma\right), \tag{2}$$

where $\boldsymbol{G}^\tau \in \{0,1\}^{N \times N}$ is the adjacency matrix for the binary GC graph of lag $\tau$ and $\odot$ denotes the Hadamard product. We further impose the following conjugate prior distributions (Miranda-Agrippino & Ricco, 2019) on $\boldsymbol{A}^\tau$: $\psi_{i,j}^\tau \sim \text{Gamma}\left(1,1\right), A_{i,j}^\tau \sim \mathcal{N}\left(0, (\psi_{i,j}^\tau)^{-1}\right)$ and on $\Sigma$: $\lambda_i \sim \text{Gamma}\left(1,1\right), \Sigma = \text{diag}\left(\lambda_1, \ldots, \lambda_N\right)^{-1}$ where $\text{diag}\left(\lambda_1, \ldots, \lambda_N\right)$ returns a matrix with its diagonal elements as $\lambda_1, \ldots, \lambda_N$. In our model, the impact of variable $j$ on $i$ is modelled by two components: $G_{i,j}^\tau \in \{0,1\}$ indicating whether there is a link between $i$ and $j$ and $A_{i,j}^\tau \in \mathbb{R}$ indicating the weight of the link. If $G_{i,j}^\tau = 0$, $j$ does not impact $i$ in lag $\tau$ regardless of the value of $A_{i,j}^\tau$ while if $G_{i,j}^\tau = 1$, $A_{i,j}^\tau$ captures the influence from $j$ to $i$. We also note that variable $j$ does not Granger-cause variable $i$ ($i,j \in \{1, \ldots, N\}$) if and only if for all $\tau$, $G_{i,j}^\tau = 0$.

### 3.1 Generalized Bernoulli Poisson Link

Before describing the model in full detail, we propose a new link function named *generalized Bernoulli Poisson link* (GBPL) that thresholds a random Poisson variable $m$ at $V \in \{1, 2, \ldots\}$ to obtain a binary variable $b$, as one of the key building blocks in our approach.

**Definition 3.1.** (Generalized Bernoulli Poisson Link)

$$m \sim \text{Poisson}\left(\gamma\right), b = \mathbf{1}(m \geq V),$$

where $\mathbf{1}(\cdot)$ is a function returning one if the condition is true and zero otherwise.

**Property 3.2.** Given $\gamma$ and $V$, one can marginalize $m$ out to get: $b \sim \text{Bernoulli}\left(1 - \sum_{v=0}^{V-1} \frac{e^{-\gamma}\gamma^v}{v!}\right)$.

*Remark.* As $b = 0$ if and only if $m < V$, $p(b = 0) = \sum_{v=0}^{V-1} p(m = v)$, thus, $p(b = 1) = 1 - \sum_{v=0}^{V-1} p(m = v)$. Moreover, as $\mathbb{E}(b) = p(b = 1)$, larger $V$ leads to lower expected probability of $b$ being one, under the same $\gamma$.

**Property 3.3.** Given $b$, the conditional posterior of $m$ conditioned on $b$, $\gamma$, and $V$ is in closed form:

$$m \sim \begin{cases} \text{TPoisson}_V\left(\gamma\right), & \text{if } b > 0 \\ \text{Categorical}_V\left([\ldots, f(v, \gamma), \ldots]\right), & \text{otherwise} \end{cases}$$

where $\text{TPoisson}_V\left(\gamma\right)$ is the Poisson distribution with parameter $\gamma$ left-truncated at $V$ (i.e., samples from that Poisson distribution are greater than or equal to $V$) and $f(v, \gamma) = \frac{\frac{e^{-\lambda}\lambda^v}{v!}}{\sum_{v'=0}^{V-1} \frac{e^{-\lambda}\lambda^{v'}}{v'!}}$ is the normalized Poisson

probability mass function. To sample from a truncated Poisson distribution, common approaches are rejection sampling (Geyer) or inverse transform sampling with inverse CDF (NumPyro). Although either approach is efficient when $V$ is large, as shown later, $V$ in our case takes a small number, leading to a relatively efficient sampling algorithm.

*Remark.* If $b > 0$, $m \geq V$ almost surely (a.s.) and one can sample $m$ from the truncated Poisson distribution at $V$ efficiently by computing the inverse Poisson cumulative distribution function (Giles, 2016). If $b = 0$, $m \in \{0, \ldots, V-1\}$ is sampled from the categorical distribution with normalized Poisson probability masses. The closed-form conditional posterior contributes to the development of an efficient algorithm of our model.

**Property 3.4.** When $V$ is set to 1, GBPL reduces to the link function proposed in Zhou (2015).

### 3.2 Poisson Factorized Granger-Causal Graph

Now we introduce our Bayesian construction on binary GC graphs with GBPL. To assist clarity, we discuss our method with only one lag, i.e., $\tau_{\max} = 1$, temporarily omitting the notation of lag $\tau$, and introduce the extension to multiple lags later.

The general idea is that we assume a binary GC graph $\boldsymbol{G} \in \{0, 1\}^{N \times N}$ is a sample of a probabilistic factorization model with $K$ latent factors: $\boldsymbol{G} \sim p(\Theta \Phi^{\mathrm{T}})$ where $\Theta \in \mathbb{R}_+^{N \times K}$ each entry of which $\theta_{i,k}$ indicates the weight of the $k^{\mathrm{th}}$ factor for variable $i$ of being a child in a GC relation and $\Phi \in \mathbb{R}_+^{N \times K}$ each entry of which $\phi_{j,k}$ indicates the weight of the $k^{\mathrm{th}}$ factor for variable $j$ of being a parent in a GC relation. In this way, whether $j$ Granger-causes $i$ depends on their interactions with all the $K$ factors: $G_{i,j} \sim p\left(\sum_{k=1}^{K} \theta_{i,k} \phi_{j,k}\right)$. Conditioned on $\Theta$ and $\Phi$, we have: $p(\boldsymbol{G}|\Theta, \Phi) = \prod_{i=1}^{N} \prod_{j=1}^{N} p(G_{i,j}|\Theta, \Phi)$, meaning that the links in $\boldsymbol{G}$ can be generated independently.

With the help of GBPL, we propose to impose the following hierarchical Bayesian prior $p(\boldsymbol{G})$:

$$r_k \sim \mathrm{Gamma}\left(1/K, 1/c\right), \quad \theta_{i,k} \sim \mathrm{Gamma}\left(a_i, 1/d_k\right), \quad \phi_{j,k} \sim \mathrm{Gamma}\left(b_j, 1/e_k\right),$$

$$M_{i,j} \sim \mathrm{Poisson}\left(\sum_{k=1}^{K} r_k \theta_{i,k} \phi_{j,k}\right), \quad G_{i,j} = \mathbf{1}(M_{i,j} \geq V), \tag{3}$$

where the first and second parameters of the gamma distribution are the shape and scale parameters, respectively; noninformative gamma priors $\mathrm{Gamma}\,(1, 1)$ are used for $a_i$, $b_j$, $d_k$, $e_k$, and $c$.

In the above model, variable $r_k$ is introduced to capture the global popularity of the $k^{\mathrm{th}}$ factor (Yang & Leskovec, 2012; 2014; Zhou, 2015). Intuitively, we can understand the vectors $\theta_{i,:}$ and $\phi_{j,:}$ as the embeddings of variables $i$ and $j$, respectively. Whether $i$ is a child of $j$ is determined by the inner product between their embeddings. Moreover, each dimension of the embeddings weights differently and $r_k$ indicates the weight of the $k^{\mathrm{th}}$ dimension.

Mathematically, the construction on $\Theta$, $\Phi$, and $\boldsymbol{r}$ can be viewed as the truncated version of a gamma process (Ferguson, 1973; Wolpert et al., 2011; Zhou, 2015) on a product space $\mathbb{R}_+ \times \Omega$: $\mathfrak{G} \sim \Gamma\mathrm{P}(\mathfrak{G}_{a,b}, 1/c)$, where $\Omega$ is a complete separable metric space, $c$ is the concentration parameter, $\mathfrak{G}_{a,b}$ is a finite and continuous base measure over $\Omega$. The corresponding Lévy measure is $\nu(drd\theta d\phi) = r^{-1}e^{-cr}dr\mathfrak{G}_{a,b}(d\theta d\phi)$. In our case, a draw from $\mathfrak{G}_{a,b}$ is a pair $(\theta_{:,k}, \phi_{:,k})$, where $\theta_{:,k} = [\theta_{1,k}, \ldots, \theta_{N,k}]$ and $\phi_{:,k} = [\phi_{1,k}, \ldots, \phi_{N,k}]$. A draw from the gamma process is a discrete distribution with countably infinite atoms from the base measure: $\mathfrak{G} = \sum_{k=1}^{\infty} r_k \delta_{\theta_{:,k}, \phi_{:,k}}$ and $r_k$ is the weight of the $k^{\mathrm{th}}$ atom. Although there are infinite atoms, the number of atoms with $r_k$ greater than $\rho \in \mathbb{R}_+$ follows $\mathrm{Poisson}(\int_\rho^\infty r^{-1}e^{-cr}dr)$ and the expectation of Poisson decreases when $\rho$ increases. In other words, the number of atoms that have relatively large weights will be finite and small, thus, a gamma process based model has an inherent shrinkage mechanism. In our case, if we set the maximum number of latent factors $K$ (i.e., the truncation level) large enough, the model will automatically learn the number of active factors.

In our model, Bayesian nonparametrics provides a principled way to build models whose complexity adapts to the data, rather than being fixed ahead of time. Even if $K$ is set to a large value, the model will not use all of the latent factors; the prior acts as a complexity penalty, preferring simpler models unless there

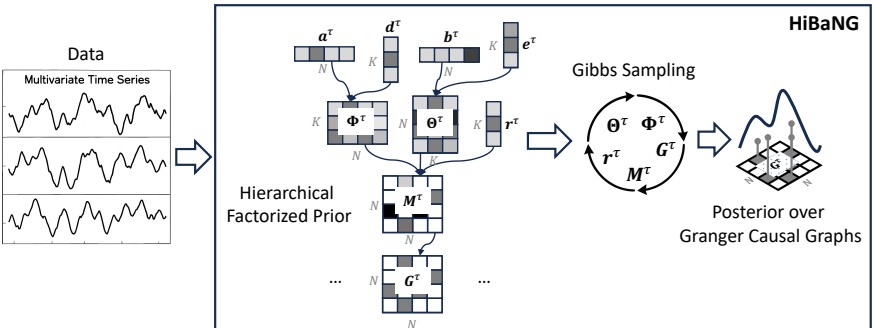

Figure 1: HiBaNG models the given MTS data via a set of binary Granger causal graphs $\boldsymbol{G}^\tau$ as latent variables with a hierarchical factorized prior.

is strong evidence to the contrary. In our model, $K$ is quite different from hyperparameters in parametric models such as the number of layers of a neural network. $K$ is a truncation level that tells the model the maximum number of latent factors it can potentially use, while the actual number it uses is determined by the data. In addition, the model will self-regularize if it uses more latent factors than necessary.

Finally, we refer to the model in Equations (2) and (3) as Poisson Factorized Granger-Causal Graph (PFGCG) and denote $\boldsymbol{G} \sim \mathrm{PFGCG}(V)$. In the case of multiple lags, we summarize the model as:

$$\boldsymbol{G}^\tau \sim \mathrm{PFGCG}^\tau(V), \quad \boldsymbol{x}_t \sim \mathcal{MN}\left(\sum_{\tau=1}^{\tau_{\max}} (\boldsymbol{A}^\tau \odot \boldsymbol{G}^\tau)\,\boldsymbol{x}_{t-\tau}, \Sigma\right), \tag{4}$$

where we have a separate generative process for the GC graph at each lag $\tau$. The overall model is shown in Figure 1.

### 3.3 Inference via Gibbs Sampling

Here we introduce how to estimate the posterior over the parameters of the above model using Gibbs sampling (Casella & George, 1992), which adheres to the detailed balance condition (Gilks et al., 1995), a fundamental property of Markov Chain Monte Carlo (MCMC) methods that guarantees the Markov chain converges to the desired posterior distribution as its stationary distribution. To further enhance efficiency, our method employs a hierarchical prior structure with conjugacy properties, with the help of several augmentation techniques between Poisson and gamma distributions (Zhou et al., 2012; Zhou, 2015). These conjugate priors ensure that all conditional distributions are analytically well-defined and computationally tractable, simplifying the sampling process. The conjugate structure not only improves computational efficiency but also contributes to the stability of Gibbs sampling, enabling the algorithm to effectively explore the posterior distribution even in challenging settings. Here we highlight the sampling of $\boldsymbol{G}^\tau$ and leave the other details in Section A.

An entry $G_{i,j}^\tau$ in $\boldsymbol{G}^\tau$ is involved in the generative process of data as in Eq. (2) and has a Bernoulli prior according to Eq. (3). Therefore, by denoting $p(G_{i,j}^\tau = 0|-) = s_{i,j}^{\tau,0}$ and $p(G_{i,j}^\tau = 1|-) = s_{i,j}^{\tau,1}$ ($-$ stands for all the other variables), we can derive:

$$s_{i,j}^{\tau,0} = \sum_{v=0}^{V-1} \frac{e^{-q_{i,j}^\tau}(q_{i,j}^\tau)^v}{v!}, \quad \text{and} \quad s_{i,j}^{\tau,1} = e^{-\frac{1}{2}\left((A_{i,j}^\tau)^2 \lambda_i U_j^\tau - 2A_{i,j}^\tau \lambda_i W_{i,j}^\tau\right)}\left(1 - s_{i,j}^{\tau,0}\right), \tag{5}$$

where:

$$q_{i,j}^\tau = \sum_{k=1}^{K} \theta_{i,k}^\tau r_k^\tau \phi_{j,k}^\tau, \quad U_j^\tau = \sum_{t=1}^{T} x_{j,t-\tau}^2, \quad W_{i,j}^\tau = \sum_{t=1}^{T} x_{i,t}^{\neg\tau,\neg j} x_{j,t-\tau},$$

$$x_{i,t}^{\neg\tau,\neg j} = x_{i,t} - \sum_{j'\neq j}^{N} A_{i,j'}^\tau G_{i,j'}^\tau x_{j',t-\tau} - \sum_{\tau'\neq\tau}^{\tau_{\max}} \sum_{j'=1}^{N} A_{i,j'}^{\tau'} G_{i,j'}^{\tau'} x_{j',t-\tau'}. \tag{6}$$

We can then sample $G_{i,j}^\tau \sim$ Bernoulli $\left( s_{i,j}^{\tau,1} / \left( s_{i,j}^{\tau,0} + s_{i,j}^{\tau,1} \right) \right)$. With the above conditional posterior, one can sample the entries of $\boldsymbol{G}^\tau$ one by one using Eq. (5). After each sample, we only need to update $W_{i,j}^\tau$ and the other statistics can be updated after all the entries are sampled.

**Computational complexity**: With the above we have that, in one Gibbs sampling iteration, the complexity of sampling $\boldsymbol{G}^\tau$ is $\mathcal{O}(N^2)$. As can be seen in Algorithm 1 in the appendix, the whole complexity of each Gibbs sampling iteration of our model is $\mathcal{O}(N^2(V + K)\tau_{\max} + T\tau_{\max}^2)$, where $N$, $T$, $V$, $K$, $\tau_{\max}$ are the number of variables, the number of timestamps, the truncation level, the number of factors, and the maximum number of lags, respectively.

### 3.4 Properties, Design Choices, and Practical Consequences

We refer to our method as HiBaNG (for hierarchical Bayesian nonparametric Granger causal discovery), which uses the PFGCG model and the Gibbs sampling algorithm described in Sections 3.2 and 3.3, respectively. Here we discuss some of HiBaNG's properties, its underlying design choices and their practical consequences.

**Separate $\boldsymbol{A}^\tau$ and $\boldsymbol{G}^\tau$ structures modeling VAR coefficients**: In our approach, $\mathbf{A} \odot \mathbf{G}$ represents coefficients, which are intrinsically sparse as $\mathbf{G}$ is binary. Thus, no sparsity-inducing penalties (Nicholson et al., 2017; Ahelegbey et al., 2016; Ghosh et al., 2018; Billio et al., 2019) or post-hoc heuristics (Nauta et al., 2019; Marcinkevičs & Vogt, 2021) are needed. Moreover, modeling binary $\mathbf{G}$ directly enables us to learn the probability of a causal link between two variables directly and one can quantify uncertainty straightforwardly.

**Independence of Granger graphs across lags:** In our model, we assume that the prior over Granger causal (GC) graphs factorizes across lags as shown in Eq. (2). Our primary motivation for this choice is to avoid imposing strong parametric or structural assumptions about temporal coherence of causal effects across lags, which may not be universally valid. While many dynamical systems exhibit smoothly decaying or persistent effects, others such as in complex observational domains such as climate or socio-economic systems can exhibit lag-specific interactions that do not conform to monotonic or smooth temporal patterns. An independent prior across lags therefore provides a more agnostic and flexible inductive bias. Importantly, although the priors are factorized, the posterior is not. The lag-specific graphs are coupled through the shared likelihood: $p(\{\boldsymbol{G}^{(\tau)}\}_{\tau=1}^{\tau_{\max}} \mid X) \propto p(X \mid \{\boldsymbol{G}^{(\tau)}\}_{\tau=1}^{\tau_{\max}}) \prod_{\tau=1}^{\tau_{\max}} p(\boldsymbol{G}^{(\tau)})$, which couples all lagged parent sets through the autoregressive structure, inducing posterior dependencies across lags when supported by the data. Thus, temporal coherence is learned rather than imposed, which is especially important in low-data regimes where strong priors can dominate inference. Although cross-lag sharing can be introduced by tying parameters or latent variables, this adds modeling commitments and weakly identified hyperparameters that can increase sensitivity to prior choices. We therefore use a factorized prior across lags, allowing coherence to emerge through the joint likelihood without additional tuning.

**Linearity and potential extension to nonlinear models:** Our model adopts a linear Gaussian VAR likelihood, which emphasizes interpretability, principled uncertainty quantification, and robustness in low-data regimes. In principle, the framework can be extended to a nonlinear likelihood of the form $\boldsymbol{x}_t = f\left(\{\boldsymbol{x}_{t-\tau}\}_{\tau=1}^{\tau_{\max}}, \{\boldsymbol{G}^\tau\}_{\tau=1}^{\tau_{\max}}\right) + \varepsilon_t$, where $f$ is a parameterized nonlinear function (e.g., a neural network). However, such an extension would generally break the conditional conjugacy structure of the proposed Bayesian model. As a result, inference would require hybrid algorithms (e.g., combining MCMC with variational or gradient-based methods), substantially increasing computational complexity and reducing robustness. We note that a linear VAR may not be a faithful generative model for many real-world systems, and that fitting a linear model to data generated by nonlinear dynamics can introduce bias. Nevertheless, one may consider using a linear model even when the underlying system is nonlinear as a pragmatic approximation. In such cases, linear Granger-based methods may still recover meaningful aspects of the causal structure when dependencies are approximately linear in expectation, or when nonlinear effects give rise to detectable linear predictive improvements. Moreover, in low-data regimes, linear models can offer more stable inference and better-calibrated uncertainty than highly flexible nonlinear alternatives.

**Suitability to low-data settings**: Our hierarchical Bayesian model is particularly suited to low-data settings because it allows for information sharing across related parameters through a structured prior. Specifically, our prior on $\boldsymbol{G}^\tau$ takes a factorized form, allowing $\theta_{i,k}$ or $\phi_{j,k}$ to capture the specific information

for an individual variable in terms of factor $k$. At a deeper level in the hierarchy, $\theta_{i,k}$ is influenced by two higher-level components: $a_i$ (capturing variable-specific traits) and $d_k$ (capturing factor-specific traits). This setup allows the model to partially pool information to learn robust estimates for $\theta_{i,k}$ even when direct data is sparse by borrowing strength from related variables (via $d_k$) and related factors (via $a_i$). As a result, the model remains expressive while enhancing generalization in low-data regimes.

**Need for the GBPL link function**: Instead of modeling binary GC graphs directly with Bernoulli distributions, we introduce GBPL as a link function that connects Bernoulli variables to underlying Poisson distributions (Zhou, 2015). This transformation enables the use of hierarchical Poisson-Gamma constructions, akin to those in Poisson matrix factorization models (Canny, 2004; Zhou et al., 2012; Gopalan et al., 2014). By leveraging the Poisson-gamma conjugacy, we gain access to a broader and more flexible set of tools for hierarchical Bayesian modeling and inference, which are difficult to apply directly to Bernoulli likelihoods.

Given the specification of the prior distributions, one can see that $\mathbb{E}\left(\sum_{k=1}^{K} r_k \theta_{i,k} \phi_{j,k}\right) = 1$ as $\mathbb{E}(\theta_{i,k}) = \mathbb{E}(\phi_{j,k}) = 1$ and $\mathbb{E}(r_k) = 1/K$. Therefore, according to Property 3.2 of GBPL, a priori, the expected sparsity of $\boldsymbol{G}$ is $N^2 \left(1 - \sum_{v=0}^{V-1} \frac{e^{-1}}{v!}\right)$. Note that $V \in \{1, 2, \dots\}$ is a hyperparameter that incorporates our prior belief of the graph sparsity, i.e., larger value of $V$ means that we encourage the model to learn sparser graphs, while the final sparsity will be determined by the model to fit the observational data. Specifically, we compute the value of $1 - \sum_{v=0}^{V-1} \frac{e^{-1}}{v!}$ with $V = \{1, 2, 3, 4\}$ as 0.2642, 0.0883, 0.0190, 0.0037, respectively. When $V = 3$, it means that less than 2% of the node pairs are expected to be connected in a graph. When $V = 4$, the sparsity is less than 0.5%, which can be overly sparse. Empirically, we observe that when $V = 4$, the sampled GC graphs from the posterior nearly have zero links, thus, we set $V \in \{1, 2, 3\}$ in practice. According to Property 3.4, the link function proposed in Zhou (2015) is a special case of GBPL (when $V = 1$). The model of Zhou (2015) cares less about the sparsity of a graph as it is given in the data. However, the expected sparsity of Zhou (2015) is 0.2642 ($V = 1$), which is too dense for many GC discovery problems. Although the expected sparsity takes finite values, the posterior graphs are not pinned to the prior grid, as data often overwhelms the prior.

**Fewer hyperparameters**: In low-data scenarios without ground-truths, selecting a model from a large set of hyperparameters is challenging. Our method has, by construction, a small number of hyperparameters as it integrates out the intermediate variables in its hierarchical Bayesian structure. Furthermore, it automatically learns $K$ from data via Bayesian nonparametrics. Indeed, the main hyperparameter of our model is $V$ that only takes a small number of discrete values, while the regularization weights in other VARs are usually continuous parameters in an infinite range.

## 4 Related Work

Here we focus on the following lines of related work in the machine learning literature and refer readers to surveys such as Shojaie & Fox (2022); Assaad et al. (2022); Gong et al. (2023) for a more comprehensive overview.

**Bayesian VARs**: have been widely used in econometrics and statistics (Breitung & Swanson, 2002; George et al., 2008; Fox et al., 2011; Nakajima & West, 2013; Ahelegbey et al., 2016; Ghosh et al., 2018; Billio et al., 2019; Ghosh et al., 2021). For comprehensive reviews, see Woźniak (2016); Miranda-Agrippino & Ricco (2019). Unlike most existing methods that focus on modeling real-valued VAR coefficients, our method models the binary GC graphs directly via a novel construction. Since BVARs are studied across multiple disciplines, comprehensive comparisons to the latest deep VARs in the same settings and datasets have not been carefully studied before.

**Deep VARs**: have recently become popular in the machine learning community (Montalto et al., 2015; Tank et al., 2018; Wang et al., 2018; Nauta et al., 2019; Khanna & Tan, 2020; Wu et al., 2020; Marcinkevičs & Vogt, 2021; Gong et al., 2022; Fan et al., 2023), where different neural network architectures or learning mechanisms have been explored (Tank et al., 2018; Nauta et al., 2019; Khanna & Tan, 2020; Marcinkevičs & Vogt, 2021; Bussmann et al., 2021; Fan et al., 2023; Zhou et al., 2024). More recently, Bayesian deep VARs have been proposed, such as ACD (Löwe et al., 2022), RHINO (Gong et al., 2022), Dyn-GFN (Tong et al., 2022),

and MCD (Varambally et al., 2024). Although these methods can also be considered Bayesian approaches, many of them have different methodologies and focuses from ours. For example, ACD (Löwe et al., 2022) focuses on discovering causal relations across samples with different underlying causal graphs but shared dynamics. RHINO (Gong et al., 2022) extends VAR by modeling instantaneous causal relations (Runge, 2020; Pamfil et al., 2020) and introducing history-dependent noise, which we do not consider in this paper. Dyn-GFN (Tong et al., 2022) is a Bayesian approach based on GFlowNets (Bengio et al., 2023) focusing on discovering causal graphs varying with time. There are also recently proposed deep VAR variants focusing on causal discovery on MTS data with missing values (Cheng et al., 2023; 2024a). Wu et al. (2024) studies a different problem from ours, where the data are event sequences and each event has an occurrence timestamp and a type. While Deep VAR models focus on expressive nonlinear dynamics and high-capacity predictors, HiBaNG adopts a different objective: learning an interpretable and uncertainty-aware posterior over Granger causal graphs using a Bayesian framework that is well-suited to low-data regimes.

**Non-VAR methods**: causal discovery on MTS data is not the focus of our paper. To capture instantaneous causal effects that are not modeled by VARs and GC, there are functional causal models such as in Hyvärinen et al. (2010); Peters et al. (2013); Pamfil et al. (2020) and methods based on dynamic Bayesian networks (DBNs) (Dean & Kanazawa, 1989; Murphy, 2002) or structured VAR models in econometrics (Swanson & Granger, 1997; Demiralp & Hoover, 2003). For DBNs, we refer readers to surveys such as Mihajlovic & Petkovic (2001); Shiguihara et al. (2021). Moreover, there are also constraint-based approaches that extend the PC algorithm (Spirtes et al., 2000) to model time-series data (Runge, 2018; Runge et al., 2019; Runge, 2020; Huang et al., 2020).

**Causal discovery for non-time-series data**: (e.g., i.i.d.) (Glymour et al., 2019) is another area with a different focus. Here we consider Bayesian methods (Lorch et al., 2021; Cundy et al., 2021; Geffner et al., 2022; Bonilla et al., 2024; Thompson et al., 2024; 2025) that model binary causal graphs as loosely related work to ours among the rich literature. These methods usually leverage gradient-based Bayesian inference algorithms such as variational inference and use reparameterization techniques (Maddison et al., 2016; Jang et al., 2016) to relax the optimization over binary causal graphs to a continuous one, while ours models binary graphs directly. Moreover, they require the discovered graphs to be directed acyclic graphs. We believe that extending their methods to time-series data with multiple lags is nontrivial.

**Other related works**: Poisson factor analysis is a generic Bayesian framework used in various areas such as graph learning (Zhou, 2015; Zhao et al., 2017), topic modeling (Zhao et al., 2018b;a), and dynamical data modeling (Schein et al., 2019). To the best of our knowledge, it has not been adapted for Granger causal discovery. The closest works to ours include Kalantari et al. (2018); Kalantari & Zhou (2021), which learn a set of latent factors from time-series data as well as a binary graph between them. There are several key differences with our method: **1)** ours is tailored to Granger causal discovery, which learns binary graphs of time-series variables instead of latent factors; **2)** ours considers multiple lags, while they only consider one; and **3)** ours is based on the proposed GBPL, while they use the original Bernoulli Poisson Link (Zhou, 2015), which is less applicable to our problem.

## 5 Experiments

Here we evaluate our method using a set of synthetic and semi-synthetic datasets as well as a real application involving the analysis of climate data. As mentioned in Section 3.4, we refer to our method as HiBaNG. In addition to the content of the main paper, we show comprehensive analyses of the empirical computational performance and convergence of HiBaNG in Sections D and F of the appendix, respectively.

### 5.1 Experimental Settings

**HiBaNG:** We use 10,000 as the maximum number of Gibbs sampling iterations, where the first 5,000 are burn-in iterations. We then collect samples from the conditional posteriors of the graphs every 10 iterations[3], which are stored in $\boldsymbol{Y} \in \mathbb{R}_+^{N \times N \times \tau_{\max} \times H}$ ($H = 500$ is the number of collections). The Bernoulli conditional posterior probability of a link between $i$ and $j$ at lag $\tau$ in collection $h \in \{1, \ldots, H\}$ is computed by Eq. (5)

---

[3]As shown in Section D, our method converges in far fewer iterations.

as $Y[i, j, \tau, h] = \frac{s_{i,j}^{\tau,1}}{s_{i,j}^{\tau,0} + s_{i,j}^{\tau,1}}$. Given the collections, to compare with other methods, we compute the averaged probability of the discovered GC graph by $\text{mean}(\max(\boldsymbol{Y}, \dim = `\tau'), \dim = `h')$ (Marcinkevičs & Vogt, 2021). As HiBaNG has an intrinsic shrinkage mechanism on $K$, we set $K = 50$, which is empirically large enough for our experiments. The only hyperparameter that we need to tune is $V$, which we vary in $\{1, 2, 3\}$.

**Baselines:** As ours is a VAR approach for GC, we mainly include baselines that are also based on the VAR framework. **1)** We compare with the widely used VAR with F-tests for Granger causality and the Benjamini–Hochberg procedure (Benjamini & Hochberg, 1995) for controlling the false discovery rate (FDR) (at $q = 0.05$), denoted as VAR (FBH) and implemented in the statsmodels library (Seabold & Statsmodels, 2010). **2)** For Bayesian methods, we compare with two classic approaches but with different prior distributions: BVAR with diffuse/noninformative priors on the coefficients, named BVAR(d), i.e., $(\{\boldsymbol{A}^\tau\}_\tau^{\tau_{\max}}, \Sigma) \propto |\Sigma|^{-\frac{N+1}{2}}$ (Litterman, 1986; Miranda-Agrippino & Ricco, 2019), whose posterior has an analytical form. BVAR with conjugate priors on the coefficients, named BVAR(c): $\psi_{i,j}^\tau \sim \text{Gamma}(1, 1), A_{i,j}^\tau \sim \mathcal{N}(0, (\psi_{i,j}^\tau)^{-1})$ and for $\Sigma$: $\lambda_i \sim \text{Gamma}(1, 1), \Sigma = \text{diag}(\lambda_1, \ldots, \lambda_N)^{-1}$. This is equivalent to an ablation of our model without $\{\boldsymbol{G}^\tau\}_\tau^{\tau_{\max}}$, for which we use Gibbs sampling with the same settings as ours. For deep/neural VARs, we compare with a method with component-wise statistical recurrent units (SRU) (Oliva et al., 2017) and its improved version (economy SRU, eSRU) (Khanna & Tan, 2020) with sample-efficient architectures. The important hyperparameters of SRU and eSRU are the strengths ($\mu_1$, $\mu_2$, $\mu_3$) of three regularization terms. We also compare with GVAR (Marcinkevičs & Vogt, 2021), which uses self-explaining neural networks (Alvarez Melis & Jaakkola, 2018) and converts the weights in the neural networks into binary GC graphs with a heuristic, stability-based procedure. As a state-of-the-art method, Jacobian Regularizer-based Neural Granger Causality (JRNGC) (Zhou et al., 2024) is also included in the comparison. For non-VAR methods, the most recent one PCMCI$^+$ (Runge, 2020) is also compared. For the baselines, we either use their original settings or follow those in Marcinkevičs & Vogt (2021), shown in Table 3. For all the compared methods, we set $\tau_{\max} = 5$ unless otherwise specified.

**Evaluation metrics:** Following Khanna & Tan (2020); Marcinkevičs & Vogt (2021), which aggregate graphs at multiple lags into one, we use four metrics to compare the discovered GC graph of a method on a dataset with the ground-truth graph. For all the baseline methods, we compute the score of a discovered GC graph from their learned VAR coefficients. For our method, the score of a GC graph is the mean of the Bernoulli posterior. We report the areas under receiver operating characteristic (AUROC) and precision-recall (AUPRC) curves by comparing the score of a discovered GC graph to the ground-truth graph. Moreover, as mentioned before, VAR (FBH) and GVAR use specific post-hoc processes to convert coefficients to binary GC graphs; thus, we also report the structural Hamming distance (SHD) between the discovered binary GC graph and the ground-truth one. Note that unlike AUCROC and AUPRC, SHD is biased toward the sparsity of the ground-truth graph; for example, for a sparse ground-truth graph, a method that always predicts no links achieves a low SHD. To measure the predictive uncertainty of the Granger-causal graphs discovered by different approaches, we report the calibration error (CE) (Guo et al., 2017), which is a widely used metric for model uncertainty and confidence (Liu et al., 2020; Kumar et al., 2019). CE examines the difference between the model's probability and the true probability given the model's output; its definition is shown in Definition 2.1 of Kumar et al. (2019). We treat the causal discovery task with $N$ variables as a binary classification problem with $N^2$ samples (i.e., predicting a Granger-causal link between a pair of variables) and then compute CE accordingly using the method in Kumar et al. (2019). For AUCROC and AUPRC, higher values indicate better performance, and for SHD and CE, lower values are better. For all the numerical results, we run our method and the baselines with 5 different random seeds and report the mean and standard deviations.

**Model selection and parameter tuning:** Importantly, we note that our task is discovering GC graphs from data without training with ground-truth, which is an unsupervised problem (e.g., akin to unsupervised clustering). As no ground-truth is given for training, we use forecasting performance, mean square error (MSE), on held-out temporal segments (last 20% of data (Gong et al., 2022)) as a proxy objective for selecting model parameters for VAR-based methods, including ours. This is consistent with how Granger causality is commonly defined—based on predictive influence. We believe this constitutes a fair comparison across all VAR-based methods, as the same procedure is used consistently. For each method, we select the parameters

of a method that give the best MSE. Our model selection is different from that of GVAR (Marcinkevičs & Vogt, 2021), where the best model is selected by comparing with the ground-truth graphs and report the best achievable performance. PCMCI$^+$ relies on conditional independence testing to infer a causal graph. Its key hyperparameter is the p-value threshold. Unlike VAR methods, PCMCI$^+$ does not perform forecasting, and thus MSE cannot be used as a model selection criterion. Therefore, we used the standard default setting of 0.05. The parameter space where we search for each method is shown in Table 3 of the appendix and the performance of our method under different parameter configurations is shown in Table 4 of the appendix.

**Setting $V$:** As stated in Section 3.4, the expected sparsity of $\mathbf{G}$ in the proposed prior distribution is $N^2 \left(1 - \sum_{v=0}^{V-1} \frac{e^{-1}}{v!}\right)$, where $V$ controls the sparsity of the graphs. We compute the value of $1 - \sum_{v=0}^{V-1} \frac{e^{-1}}{v!}$ with $V \in \{1, 2, 3, 4\}$ as 0.2642, 0.0883, 0.0190, 0.0037, respectively. Thus, setting $V = 3$ already induces significant sparsity, which matches the empirical needs of our experiments. We see that when $V = 4$, the sampled GC graphs from the posterior nearly have zero links, thus, we set $V \in \{1, 2, 3\}$ in practice.

**Setting $K$:** We believe that $K = 50$ is sufficiently large in our experimental settings. This can be seen on the RHS of Figure 2, which shows $r_k^\tau$ for all lags and all $K$ factors as a $K \times \tau_{\max}$ matrix ($K = 50$ and $\tau_{\max} = 5$). Recall that $r_k^\tau$ indicates the weight of latent factor $k$ at lag $\tau$. If we look at one column of the figure that shows the weights of the 50 factors for one lag, we can see that only a few entries have large values, meaning that only a few factors are active among $K = 50$. We had similar observations in other datasets as well.

## 5.2 Synthetic and Semi-synthetic Datasets

We report the results on toy synthetic datasets in Section B. For more comprehensive quantitative comparisons, we conduct our experiments on three widely used benchmark datasets, detailed as follows.

**Lorenz 96** (Lorenz, 1996) is a standard benchmark synthetic MTS dataset for GC, which is generated from the following nonlinear differential equations: $\frac{dx_{i,t}}{dt} = (x_{i+1,t} - x_{i-2,t})x_{i-1,t} - x_{i,t} + F$, for $1 \leq i \leq N$, where $F$ is a constant that models the magnitude of the external forcing. The system dynamics become increasingly chaotic for higher values of $F$ (Karimi & Paul, 2010). We set $N = 40$, $F = 40$, and $T = \{100, 500\}$, which mimic noisy observations with reasonably large numbers of variables but few observations.

Following Marcinkevičs & Vogt (2021), we evaluate the methods on another synthetic dataset generated by the **Lotka–Volterra** model (Bacaër & Bacaër, 2011), where we use $N = 40$ and $T = \{200, 500\}$. For the other parameters of the Lotka–Volterra model, we use the same settings as in Marcinkevičs & Vogt (2021).

Finally, we consider the **FMRI** dataset with realistic simulations of blood-oxygen-level dependent (BOLD) time series (Smith et al., 2011). In the dataset, multivariate BOLD time series are generated from a known directed network structure via a biophysically grounded forward model that links neural activity to hemodynamic responses. Specifically, the simulator uses Dynamic Causal Modeling (DCM)–style architectures (Friston et al., 2003) to produce realistic BOLD signals from a predefined connectivity matrix. The corresponding directed adjacency matrix used in the generation process is treated as the ground truth when evaluating causal discovery methods. Following Khanna & Tan (2020); Marcinkevičs & Vogt (2021), we use 5 replicates from the simulation no. 3 of the original dataset, where $N = 15$ and $T = 200$ are pre-specified as standard settings. We introduce a straightforward metric to measure the severity of low-data conditions in a dataset: $\beta = N/T$, meaning that larger $\beta$ indicates a dataset has a larger number of variables with a smaller number of observations. We show the value of $\beta$ for each dataset in Table 1. We notice that Cheng et al. (2024b) recently introduced a few new benchmark datasets with many timestamps (e.g., from 8,000 to 50,000). Our focus is on low-data regimes with fewer timestamps, and their ground-truth graphs are undirected (i.e., the adjacency matrices are symmetric), while our method discovers directed graphs; therefore, these datasets are less applicable to our problem.

We present the results for AUCROC and AUPRC in Table 1. Overall, our proposed method, HiBaNG, demonstrates superior performance across most datasets and metrics. A notable observation is the enhanced performance of HiBaNG as the parameter $\beta$ increases. This trend underscores the robustness of our model in severe low-data situations, where traditional methods often struggle. For the FMRI dataset, where the severity of low-data conditions is reduced, HiBaNG ranks second in both AUCROC and AUPRC. This result aligns

Table 1: AUCROC and AUPRC results on semi-synthetic datasets. VAR (FBH) and BVAR(d) failed to learn when $T = 100$ on Lorenz 96 and $T = 200$ on Lotka–Volterra. Best and second best results are highlighted in boldface and underline texts, respectively. $\beta = N/T$ indicates the ratio of the number of variables to the number of observations.

| | Lorenz 96 $T = 100, \beta = 0.4$ | | Lorenz 96 $T = 500, \beta = 0.08$ | | Lotka–Volterra $T = 200, \beta = 0.2$ | | Lotka–Volterra $T = 500, \beta = 0.08$ | | FMRI $\beta = 0.075$ | |
|---|---|---|---|---|---|---|---|---|---|---|
| | AUCROC↑ | AUPRC↑ | AUCROC↑ | AUPRC↑ | AUCROC↑ | AUPRC↑ | AUCROC↑ | AUPRC↑ | AUCROC↑ | AUPRC↑ |
| BVAR(c) | 0.47±0.02 | 0.09±0.01 | 0.73±0.02 | 0.43±0.02 | 0.50±0.01 | 0.50±0.011 | 0.78±0.04 | 0.52±0.05 | 0.66±0.08 | 0.42±0.12 |
| BVAR(d) | - | - | 0.73±0.03 | 0.43±0.02 | - | - | 0.67±0.01 | 0.23.±0.03 | 0.68±0.06 | 0.40±0.06 |
| VAR (FBH) | - | - | 0.72±0.01 | 0.39±0.03 | - | - | 0.68±0.03 | 0.18±0.01 | 0.60±0.04 | 0.32±0.02 |
| PCMCI$^+$ | 0.62±0.02 | 0.17±0.01 | 0.82±0.02 | 0.59±0.02 | 0.72±0.03 | 0.44±0.02 | 0.78±0.01 | 0.47±0.01 | **0.89**±0.04 | **0.67**±0.07 |
| SRU | 0.53±0.01 | 0.12±0.01 | 0.82±0.03 | 0.57±0.04 | 0.55±0.02 | 0.54±0.01 | 0.61±0.02 | 0.32±0.02 | 0.66±0.02 | 0.32±0.03 |
| eSRU | 0.54±0.03 | 0.12±0.01 | 0.84±0.01 | 0.63±0.05 | 0.64±0.03 | 0.58±0.01 | 0.67±0.03 | 0.36±0.01 | 0.72±0.01 | 0.47±0.01 |
| GVAR | 0.57±0.01 | 0.15±0.03 | 0.83±0.01 | 0.63±0.01 | 0.66±0.02 | 0.62±0.01 | 0.81±0.03 | 0.61±0.04 | 0.72±0.02 | 0.57±0.06 |
| JRNGC | 0.58±0.07 | 0.14±0.04 | 0.79±0.01 | 0.55±0.03 | 0.68±0.02 | 0.32±0.03 | **0.87**±0.02 | 0.61±0.03 | 0.69±0.03 | 0.45±0.04 |
| HiBaNG | **0.71**±0.03 | **0.35**±0.04 | **0.86**±0.02 | **0.68**±0.02 | **0.73**±0.01 | **0.73**±0.01 | 0.84±0.03 | **0.66**±0.03 | 0.73±0.03 | 0.60±0.03 |

with expectations, as richer datasets reduce the relative advantage of our Bayesian approach. Comparing HiBaNG with BVAR(c), which essentially represents HiBaNG without the integration of binary GC graphs, reveals that HiBaNG consistently outperforms BVAR(c). This comparison highlights the significance of incorporating binary GC graphs into the Bayesian framework, facilitating a clearer and more interpretable understanding of causal links while enhancing predictive accuracy.

The results for SHD and CE are displayed in Table 2. For SHD, we include methods capable of converting their coefficients into sparse graphs, while for CE, we concentrate on the top-performing methods based on AUCROC and AUPRC scores. It is important to note that SHD is influenced by the method's approach to generating binary graphs, which may introduce bias depending on the sparsity of the ground-truth graphs. Our method's capability to directly sample binary graphs without relying on arbitrary thresholds provides a distinct advantage. Regarding CE, our method achieves the lowest error rates across almost all datasets. This indicates that HiBaNG's predictive confidence is highly aligned with its accuracy, a result of its inherent uncertainty-aware design. Such alignment is crucial in real-world applications, where understanding the reliability of causal predictions can inform better decision-making.

Table 2: SHD and CE on semi-synthetic datasets.

| | Lorenz 96 $T = 100$ | Lorenz 96 $T = 500$ | Lotka–Volterr $T = 200$ | Lotka–Volterr $T = 500$ | FMRI |
|---|---|---|---|---|---|
| | | | SHD↓ | | |
| VAR (FBH) | - | 98.40±2.4 | - | 74.20±10.4 | 28.8±1.3 |
| PCMCI$^+$ | 405.0±15.0 | 411.0±5.0 | 494.0±20.0 | 423.0±20.0 | 70.00±2.00 |
| GVAR | 389.6±220.6 | 127.4±76.8 | 279.0±104.1 | 82.8±24.1 | 71.6±21.8 |
| HiBaNG | **117.7**±3.3 | **71.8**±4.0 | **67.0**±2.4 | **45.0**±4.4 | **24.2**±0.7 |
| | | | CE↓ | | |
| BVAR (d) | - | 0.10±0.01 | - | 0.11±0.01 | 0.11±0.02 |
| PCMCI$^+$ | 0.25±0.01 | 0.27±0.01 | 0.27±0.01 | 0.28±0.01 | 0.31±0.02 |
| GVAR | 0.07±0.01 | 0.15±0.01 | 0.08±0.01 | 0.10±0.01 | 0.19±0.01 |
| JRNGC | **0.01**±0.01 | 0.09±0.01 | 0.07±0.01 | 0.13±0.01 | 0.09±0.02 |
| HiBaNG | 0.11±0.01 | **0.08**±0.01 | **0.05**±0.01 | **0.04**±0.01 | **0.07**±0.01 |

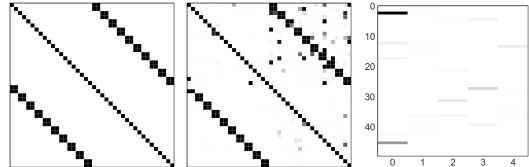

Figure 2: Qualitative analysis of HiBaNG on Lotka–Volterra. Left: Ground-truth GC graph, middle: Bernoulli posterior mean of the discovered GC graphs, right: Matrix of $\{r_k^\tau\}_{k=1,\tau}^{K,\tau_{\max}}$ showing that $K = 50$ is sufficiently large in our experiments (see Section 5.1 for more details). Each rectangle indicates a value of a matrix and brighter colors indicates larger values.

In Figure 2, we compare the Bernoulli posterior mean of HiBaNG with the ground-truth graph on Lotka–Volterra. We can see that the posterior mean discovered by our method is well aligned with the ground-truth graph, where brighter rectangles indicate a higher probability of a GC link between two variables. Finally, recall that $r_k^\tau$ in Eq. (3) models the weight of latent factor $k$ at lag $\tau$. We plot $\{r_k^\tau\}_{k=1,\tau}^{K,\tau_{\max}}$ as a $K \times \tau_{\max}$ matrix. The matrix is quite sparse: only a few entries have large values, and only a few factors are active among $K = 50$. This demonstrates the shrinkage mechanism on $K$.

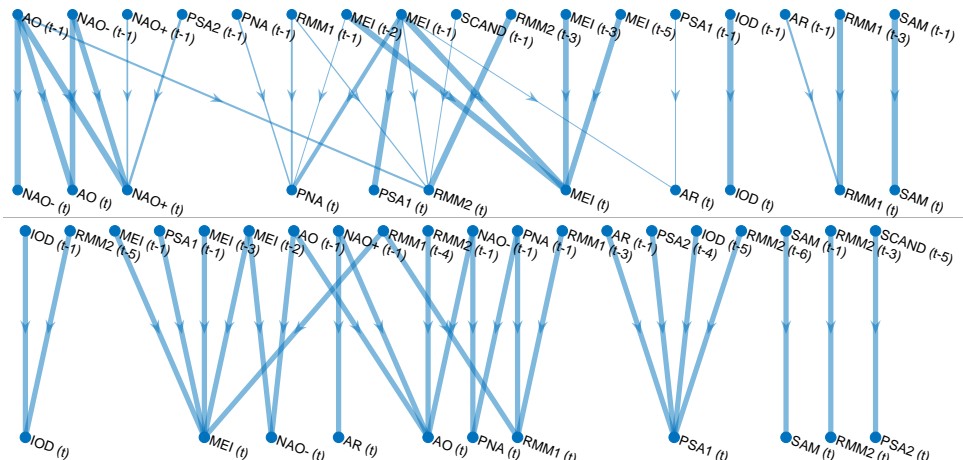

Figure 3: Results on JR55. Up: HiBaNG discovered causal links between indices on JR55 where the weights are from the Bernoulli posterior of the graph (links with weights less than 0.2 are not shown) and thicker links indicate stronger connections. Down: PCMCI$^+$ discovered causal links.

### 5.3 Qualitative Analysis on Climate Reanalysis Data

We qualitatively analyze our method's performance on climate data obtained from the Japanese reanalysis of the atmosphere (JRA55) (Kobayashi et al., 2015), detailed in Section C. We compare HiBaNG and PCMCI$^+$ with lag $\tau_{\max} = 6$ in Figure 3. We qualitatively explain the physical interpretability of relationships discovered by PCMCI and HiBaNG: 1) ENSO (MEI) autocorrelation: HiBaNG captures autocorrelations in MEI extending back to $t$–5, aligning with the known persistence of ENSO over approximately 6 months, as discussed in Harries & O'Kane (2021) (Figure 5). PCMCI only captures MEI autocorrelations up to $t$–3, suggesting a more limited temporal sensitivity. 2) MEI→PSA1 directionality: HiBaNG correctly infers MEI($t$–1)→PSA1($t$), consistent with the established ENSO-to-PSA1 influence via modulation of midlatitude tropospheric flow (O'Kane & Franzke, 2025). PCMCI infers the reverse direction (PSA1($t$–1)→MEI($t$)). 3) PNA links: Both methods identify the PNA autocorrelation at a one-month lag, consistent with observed behavior. However, HiBaNG additionally identifies the MEI($t$–1)→PNA($t$) edge, also seen in Harries & O'Kane (2021) (Figure 9b), supporting known teleconnections. PCMCI instead finds NAO–($t$–1)→PNA($t$), which is plausible but not as directly supported by prior work. 4) Northern Hemisphere modes: HiBaNG provides a more complete representation of the interconnected dynamics among AO, NAO+, NAO–, and AR. For example, the HiBaNG graph includes: PSA2($t$–1)→NAO+($t$); PSA1($t$)→AR($t$); and NAO+ and NAO– autocorrelations at $t$–1. These are consistent with Harries & O'Kane (2021) (Figures 6 and 7), and are not captured by PCMCI. Overall, the HiBaNG graph shows a larger and more diverse set of edges consistent with those previously inferred from the JRA55 reanalysis via Bayesian structure learning as reported in Harries & O'Kane (2021). We note that this experiment is intended primarily as a qualitative demonstration of our framework's interpretability and practical applicability, with PCMCI serving as a reference. We do not claim that HiBaNG should systematically outperform PCMCI on this task, particularly since PCMCI is more expressive in principle due to its support for nonlinear conditional independence tests.

## 6 Conclusion

We have presented a novel Bayesian VAR model tailored to Granger causal discovery on MTS data in low-data settings. Our method leverages a hierarchical Bayesian framework that separates Granger causal relationships into binary causal graphs and real-valued weights. Through extensive experiments on synthetic, semi-synthetic, and real-world datasets, we have demonstrated that our approach can perform better in low-data regimes. Regarding limitations, our method is based on the Granger causality framework, which assumes that causal relationships are reflected in time-lagged dependencies. While this assumption is appropriate for many applications, it may not hold universally, and practitioners should exercise caution when interpreting the results.

## Broader Impact Statement

This paper introduces a novel method for discovering causal relationships from observational data, grounded in the framework of Granger causality and a factorized representation of causal structure. By leveraging these assumptions, the method enables scalable and interpretable causal discovery, which can benefit applications in fields such as economics, neuroscience, and climate science, where temporal data is abundant. The approach opens up new possibilities for causal inference in high-dimensional settings, providing a foundation for future work that can relax or adapt these assumptions to broader domains.

Because the method relies on assumptions such as Granger causality and factorized causal structures, there is a risk of drawing incorrect conclusions if these assumptions do not hold. Misinterpretation or misuse of inferred causal relationships could lead to flawed decisions or reinforce biases present in the data, especially in high-stakes domains.

### Acknowledgments

The authors thank the reviewers and action editor for the insightful reviews.

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

## A   Inference via Gibbs Sampling

**Sampling $\boldsymbol{A}^\tau$**   With the conjugacy of normal distributions, one can sample the entries of $\boldsymbol{A}^\tau$ one by one by:

$$A_{i,j}^\tau \sim \begin{cases} \mathcal{N}\left(0, (\psi_{i,j}^\tau)^{-1}\right), & \text{if } G_{i,j}^\tau = 0 \\ \mathcal{N}\left(\mu_{i,j}^\tau, \sigma_{i,j}^\tau\right), & \text{otherwise} \end{cases} \tag{7}$$

where:

$$\sigma_{i,j}^{\tau} = \left( \lambda_i G_{i,j}^{\tau} \sum_{t=1}^{T} x_{j,t-\tau}^2 + \psi_{i,j}^{\tau} \right)^{-1}, \tag{8}$$

$$\mu_{i,j}^{\tau} = \sigma_{i,j}^{\tau} G_{i,j}^{\tau} \lambda_i \left( \sum_{t=1}^{T} x_{i,t}^{\neg\tau,\neg j} x_{j,t-\tau} \right), \tag{9}$$

$$x_{i,t}^{\neg\tau,\neg j} = x_{i,t} - \sum_{j'\neq j}^{N} A_{i,j'}^{\tau} G_{i,j'}^{\tau} x_{j',t-\tau} - \sum_{\tau'\neq\tau}^{\tau_{\max}} \sum_{j'=1}^{N} A_{i,j'}^{\tau'} G_{i,j'}^{\tau'} x_{j',t-\tau'} \tag{10}$$

**Sampling $\psi_{i,j}^{\tau}$ and $\lambda_i$**   With the conjugacy between normal and gamma distributions, one can sample $\psi_{i,j}^{\tau}$ and $\lambda_i$ from their conditional gamma posteriors:

$$\psi_{i,j}^{\tau} \sim \text{Gamma}\left(1.5, 1/(A_{i,j}^{\tau}/2 + 1)\right), \tag{11}$$

$$\lambda_i \sim \text{Gamma}\left(1 + T/2, \left(1 + \sum_{t=1}^{T}(x_{i,t} - \sum_{\tau=1}^{\tau_{\max}} \sum_{j=1}^{N} A_{i,j}^{\tau} G_{i,j}^{\tau} x_{j,t-\tau})\right)^{-1}\right). \tag{12}$$

**Sampling $M^{\tau}$**   With GBPL, we can sample:

$$M_{i,j}^{\tau} \sim \begin{cases} \text{Categorical}_V\left(\left[\ldots, \dfrac{\frac{e^{-q_{i,j}^{\tau}}(q_{i,j}^{\tau})^v}{v!}}{\sum_{v'=0}^{V} \frac{e^{-q_{i,j}^{\tau}}(q_{i,j}^{\tau})^{v'}}{v'!}}, \ldots\right]\right), & \text{if } G_{i,j}^{\tau} = 0 \\ \text{TPoisson}_V\left(q_{i,j}^{\tau}\right), & \text{otherwise} \end{cases} \tag{13}$$

where $q_{i,j}^{\tau} = \sum_{k=1}^{K} \theta_{i,k}^{\tau} r_k^{\tau} \phi_{j,k}^{\tau}$.

**Sampling $M_{i,j,k}^{\tau}$**   With the relationships between Poisson and multinomial distributions, we can sample:

$$[\ldots, m_{i,j,k}^{\tau}, \ldots] \sim \text{Multinomial}_K\left(m_{i,j}^{\tau}; \left[\ldots, \frac{q_{i,j,k}^{\tau}}{\sum_{k'=1}^{K} q_{i,j,k'}^{\tau}}, \ldots\right]\right), \tag{14}$$

where $q_{i,j,k}^{\tau} = \theta_{i,k}^{\tau} r_k^{\tau} \phi_{j,k}^{\tau}$.

**Sampling $\theta_{i,k}^{\tau}$, $\phi_{j,k}^{\tau}$, and $r_k^{\tau}$**

$$\theta_{i,k}^{\tau} \sim \text{Gamma}\left(a_i^{\tau} + \sum_{j=1}^{N} M_{i,j,k}^{\tau}, \frac{1}{d_k^{\tau} + r_k^{\tau} \sum_{j=1}^{N} \phi_{j,k}^{\tau}}\right), \tag{15}$$

$$\phi_{j,k}^{\tau} \sim \text{Gamma}\left(b_j^{\tau} + \sum_{i=1}^{N} M_{i,j,k}^{\tau}, \frac{1}{e_k^{\tau} + r_k^{\tau} \sum_{i=1}^{N} \theta_{i,k}^{\tau}}\right), \tag{16}$$

$$r_k^{\tau} \sim \text{Gamma}\left(1/K + \sum_{i=1,j=1}^{N} M_{i,j,k}^{\tau}, \frac{1}{c^{\tau} + \sum_{i=1,j=1}^{N} \theta_{i,k}^{\tau} \phi_{j,k}^{\tau}}\right). \tag{17}$$

Table 3: Hyperparameter settings.

| Model | $\tau_{\max}$ | # hidden layers | # hidden units | # training epochs | Learning rate | Mini-batch size | Parameter space |
|---|---|---|---|---|---|---|---|
| VAR (FBH) | {1,3,5} | NA | NA | NA | NA | NA | NA |
| BVAR(d) | {1,3,5} | NA | NA | NA | NA | NA | NA |
| BVAR(c) | {1,3,5} | NA | NA | NA | NA | NA | NA |
| SRU | NA | 1 | 10 | 2000 | 1.0e-3 | 50 | $\mu_1 = [0.01, 0.05]$ $\mu_2 = [0.01, 0.05]$ $\mu_3 = [0.01, 1.0]$ |
| eSRU | NA | 2 | 10 | 2000 | 1.0e-3 | 50 | $\mu_1 = [0.01, 0.05]$ $\mu_2 = [0.01, 0.05]$ $\mu_3 = [0.01, 1.0]$ |
| GVAR | {1,3,5} | 2 | 50 | 1,000 | 1.0e-4 | 64 | $\lambda = [0.0, 3.0]$ $\gamma = [0.0, 0.1]$ |
| HiBaNG | {1,3,5} | NA | NA | 10,000 | NA | NA | $V = \{1, 2, 3\}$ |

**Sampling $a_i^\tau$ and $b_j^\tau$**  By introducing auxiliary variables from the Chinese Restaurant Table (CRT) distribution (Zhou et al., 2012; Zhou & Carin, 2013), we can sample:

$$l_{i,k}^\tau \sim \mathrm{CRT}\left(\sum_{j=1}^{N} M_{i,j,k}^\tau, a_i^\tau\right), \tag{18}$$

$$a_i^\tau \sim \mathrm{Gamma}\left(1 + \sum_{k=1}^{K} l_{i,k}^\tau, \frac{1}{1 + \sum_{k=1}^{K} \log(1 + r_k^\tau \sum_{j=1}^{N} \phi_{j,k}^\tau / d_k^\tau)}\right), \tag{19}$$

$$o_{j,k}^\tau \sim \mathrm{CRT}\left(\sum_{i=1}^{N} M_{i,j,k}^\tau, b_j^\tau\right), \tag{20}$$

$$b_j^\tau \sim \mathrm{Gamma}\left(1 + \sum_{k=1}^{K} o_{j,k}^\tau, \frac{1}{1 + \sum_{k=1}^{K} \log(1 + r_k^\tau \sum_{i=1}^{N} \theta_{i,k}^\tau / e_k^\tau)}\right). \tag{21}$$

**Sampling $d_k^\tau$, $e_k^\tau$ and $c^\tau$**

$$d_k^\tau \sim \mathrm{Gamma}\left(\sum_{i=1}^{N} a_i^\tau + 1, \frac{1}{\sum_{i=1}^{N} \theta_{i,k}^\tau + 1}\right), \tag{22}$$

$$e_k^\tau \sim \mathrm{Gamma}\left(\sum_{j=1}^{N} b_j^\tau + 1, \frac{1}{\sum_{i=1}^{N} \phi_{j,k}^\tau + 1}\right), \tag{23}$$

$$c^\tau \sim \mathrm{Gamma}\left(2, \frac{1}{\sum_{k=1}^{K} r_k^\tau + 1}\right). \tag{24}$$

## B  Synthetic Data

We conduct experiments on synthetic data generated from a VAR model specified in Eq. (1) to test whether our method can discover the ground-truth graphs. Given $N = 16$ and $\tau_{\max} = 6$, we construct $\{\boldsymbol{A}^\tau\}_{\tau=1}^{\tau_{\max}}$ by first specifying the nonzero entries (i.e., the ground-truth causal graphs) and then, for each nonzero entry, sampling $A_{i,j}^\tau \sim \mathrm{Uniform}(0.1, 0.2)$. We then generate $T = 1,000$ samples, initialize $\boldsymbol{x}_0$ from a standard normal distribution, and sample $\epsilon_t \sim \mathcal{N}(0, 0.01)$. We show the results of our method in Figure 4, where we also fit a randomly initialized VAR (Seabold & Statsmodels, 2010) to the data as a reference. The ground-truth graphs at different lags exhibit diverse patterns, and our method recovers them well (also reflected by higher AUROC and AUPRC). Unlike VAR, our method directly infers binary graphs without using thresholds or hypothesis tests.

---

**Algorithm 1:** Inference Algorithm for HiBaNG.

---

**input** : MTS data $\boldsymbol{X}$, number of lags $\tau_{\mathrm{max}}$, hyperparameter $V$
**output**: Posterior samples of $\{\boldsymbol{A}^{\tau}\}_{\tau}^{\tau_{\mathrm{max}}}$ and $\{\boldsymbol{G}^{\tau}\}_{\tau}^{\tau_{\mathrm{max}}}$
Initialize all the variables;
**while** *Not converged* **do**
    **for** $i = 1 \dots N$ **do**
        | Sample $\lambda_i$;
    **end**
    **for** $\tau = 1 \dots \tau_{max}$ **do**
        **for** $i = 1 \dots N, \; j = 1 \dots N$ **do**
            | Sample $M_{i,j}^{\tau}$ and $M_{i,j,k}^{\tau}$;
        **end**
        Sample $c^{\tau}$;
        **for** $i = 1 \dots N$ **do**
            | Sample $a_i^{\tau}$ and $b_i^{\tau}$;
        **end**
        **for** $k = 1 \dots K$ **do**
            | Sample $d_k^{\tau}, e_k^{\tau}, r_k^{\tau}$;
        **end**
        **for** $i = 1 \dots N$ **do**
            **for** $k = 1 \dots K$ **do**
                | Sample $\theta_{i,k}^{\tau}$ and $\phi_{i,k}^{\tau}$;
            **end**
        **end**
        **for** $i = 1 \dots N$ **do**
            **for** $j = 1 \dots N$ **do**
                | Sample $A_{i,j}^{\tau}, \psi_{i,j}^{\tau}, G_{i,j}^{\tau}$;
            **end**
        **end**
    **end**
**end**

---

## C   Further Introduction to the Climate Reanalysis Data

Following Harries & O'Kane (2021), we compute 13 indices that diagnose the activity of the major atmospheric, tropospheric, and convective global climate modes at monthly resolution from 1960 to 2005, resulting in an MTS dataset with $N = 13$ and $T = 551$. The climate indices' names are shown in Table 7. Among the indices, the Multivariate El Niño–Southern Oscillation Index (MEI) is representative in that its time series is associated with regionally distributed, coherent responses in the atmosphere and surface ocean. In Figure 5(a), we plot the MEI time series as an example; it characterizes the El Niño/La Niña cycle. Positive values of the MEI are associated with El Niño periods, whereas negative values are associated with La Niña periods; the magnitude of the index is proportional to the strength of the event. For example, according to the MEI, there was a strong El Niño in April 1998. Figure 5(b) illustrates how much warmer (red) and cooler (blue) the surface air temperature was in that month with respect to the average April. This map shows a large warm patch over the eastern Pacific Ocean, which is typical of El Niño. Conversely, the MEI indicates that August 1988 featured a strong La Niña; the associated surface air temperature map is shown in Figure 5(c). This map illustrates how different the surface air temperature was with respect to an average August. As is typical of La Niña events, the eastern Pacific region is anomalously cool.

Table 4: Performance of HiBaNG on FMRI in different parameters.

| $\tau_{\max}$ | $V$ | $K$ | AUROC | AUPRC |
|---|---|---|---|---|
| 1 | 1 | 5 | $0.711 \pm 0.044$ | $0.550 \pm 0.037$ |
| 1 | 1 | 20 | $0.725 \pm 0.046$ | $0.552 \pm 0.034$ |
| 1 | 1 | 50 | $0.714 \pm 0.041$ | $0.550 \pm 0.035$ |
| 1 | 2 | 5 | $0.715 \pm 0.051$ | $0.544 \pm 0.037$ |
| 1 | 2 | 20 | $0.713 \pm 0.052$ | $0.543 \pm 0.036$ |
| 1 | 2 | 50 | $0.712 \pm 0.048$ | $0.543 \pm 0.033$ |
| 1 | 3 | 5 | $0.705 \pm 0.058$ | $0.539 \pm 0.035$ |
| 1 | 3 | 20 | $0.710 \pm 0.060$ | $0.539 \pm 0.038$ |
| 1 | 3 | 50 | $0.706 \pm 0.044$ | $0.538 \pm 0.034$ |
| 3 | 1 | 5 | $0.737 \pm 0.039$ | $0.590 \pm 0.038$ |
| 3 | 1 | 20 | $0.731 \pm 0.032$ | $0.585 \pm 0.037$ |
| 3 | 1 | 50 | $0.739 \pm 0.036$ | $0.591 \pm 0.038$ |
| 3 | 2 | 5 | $0.732 \pm 0.031$ | $0.583 \pm 0.039$ |
| 3 | 2 | 20 | $0.728 \pm 0.012$ | $0.597 \pm 0.014$ |
| 3 | 2 | 50 | $0.724 \pm 0.036$ | $0.584 \pm 0.044$ |
| 3 | 3 | 5 | $0.723 \pm 0.028$ | $0.578 \pm 0.040$ |
| 3 | 3 | 20 | $0.731 \pm 0.034$ | $0.582 \pm 0.042$ |
| 3 | 3 | 50 | $0.736 \pm 0.034$ | $0.585 \pm 0.044$ |
| 5 | 1 | 5 | $0.736 \pm 0.022$ | $0.589 \pm 0.023$ |
| 5 | 1 | 20 | $0.742 \pm 0.029$ | $0.594 \pm 0.026$ |
| 5 | 1 | 50 | $0.742 \pm 0.030$ | $0.593 \pm 0.027$ |
| 5 | 2 | 5 | $0.732 \pm 0.026$ | $0.587 \pm 0.033$ |
| 5 | 2 | 20 | $0.743 \pm 0.030$ | $0.592 \pm 0.031$ |
| 5 | 2 | 50 | $0.740 \pm 0.029$ | $0.591 \pm 0.029$ |
| 5 | 3 | 5 | $0.733 \pm 0.017$ | $0.583 \pm 0.031$ |
| 5 | 3 | 20 | $0.738 \pm 0.030$ | $0.590 \pm 0.032$ |
| 5 | 3 | 50 | $0.734 \pm 0.023$ | $0.589 \pm 0.035$ |

## D    Empirical Computational Performance

In addition to the complexity analysis in Section 3.3, we empirically study computational efficiency by examining two aspects: the running time of a single Gibbs sampling iteration and the number of iterations required for convergence. For all experiments in this section, we ran our method on an Apple laptop with an M1 Pro processor. We report the running time (seconds per Gibbs sampling iteration) as we vary the number of variables and time steps on Lorenz 96 in Figure 6c. Inference is efficient and scales gracefully with $T$. Figure 6 shows HiBaNG's MSE over iterations on Lorenz 96. Although we set the maximum number of Gibbs sampling iterations to 10,000, in most cases our method converges in around 200 iterations. We also observe that in low-data regimes (e.g., $T = 100$ on Lorenz 96), samples from our model may exhibit higher variance early in inference, as expected with fewer time steps. In general, our method converges within half an hour on a laptop in most cases.

Table 5: SHD for VAR (FBH), GVAR, and HiBaNG with $\tau_{\max} = 1$. Means and standard deviations are computed over 5 replicates on each dataset.

|  | L96 | | LV | | FMRI |
|---|---|---|---|---|---|
|  | $T = 100$ | $T = 500$ | $T = 200$ | $T = 500$ |  |
| VAR (FBH) | - | 72.00±4.69 | 125.60±21.88 | 451.40±69.85 | 26.00±1.10 |
| GVAR | 373.20±39.77 | 93.60±55.23 | 147.80±80.36 | 111.00±39.75 | 51.00±16.79 |
| HiBaNG | 112.91±1.95 | 71.43±3.52 | 186.13±21.84 | 49.73±4.72 | 25.17±1.43 |

Table 6: SHD for VAR (FBH), GVAR, and HiBaNG with $\tau_{\max} = 3$. Means and standard deviations are computed over 5 replicates on each dataset.

|  | L96 | | LV | | FMRI |
|---|---|---|---|---|---|
|  | $T = 100$ | $T = 500$ | $T = 200$ | $T = 500$ |  |
| VAR (FBH) | - | 79.60±3.32 | 84.40±4.18 | 75.40±5.85 | 25.80±1.33 |
| GVAR | 577.40±47.51 | 133.20±107.22 | 213.80±117.98 | 128.60±68.31 | 50.00±18.99 |
| HiBaNG | 114.75±1.68 | 73.18±2.67 | 75.30±9.21 | 45.50±2.34 | 24.78±1.08 |

Table 7: Climate index names.

| Index name | Abbreviation |
|---|---|
| Atlantic Oscillation | AO |
| Indian Ocean Dipole | IOD |
| Multivariate El Niño Southern Oscillation Index | MEI |
| North Atlantic Oscillation (positive and negative phases) | NAO+/- |
| Atlantic Ridge patterns | AR |
| Scandinavian blocking patterns | SCAND |
| Pacific North American patterns | PNA |
| Pacific South American patterns | PSA1/2 |
| Southern Annular Mode | SAM |
| Wheeler-Hendon Madden-Julian oscillation | RMM1/2 |

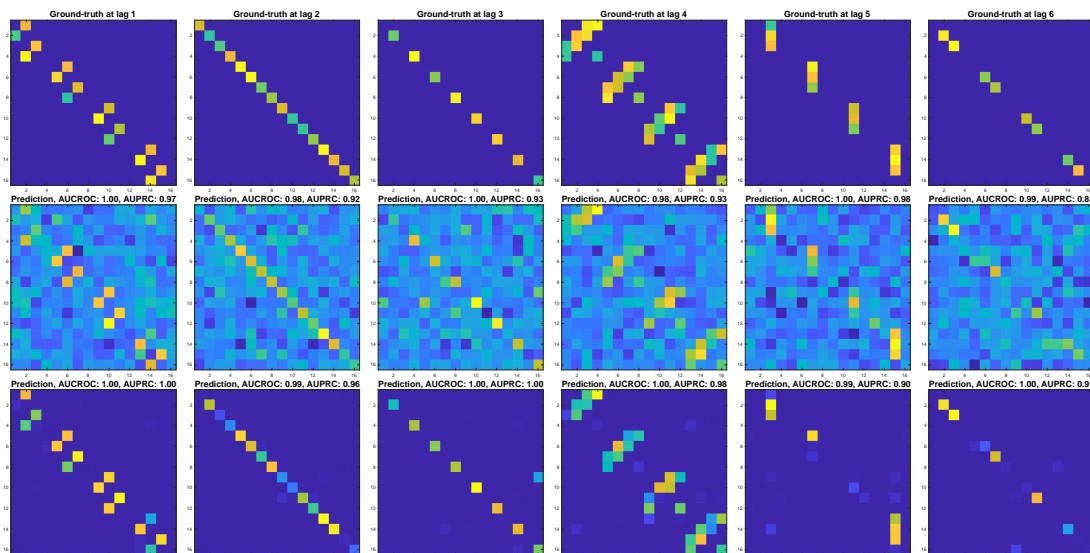

Figure 4: Synthetic dataset. First to third rows: ground-truth graphs, coefficients of VAR, and GC graphs of ours at different lags. Columns: $\tau = 1, \ldots, 6$. The AUCROC and AUPRC scores at each lag of VAR and ours are shown in the sub-captions. Mean AUCROC over all the lags is 0.99 (VAR) and 1.0 (ours); Mean AUPRC is 0.93 (VAR) and 0.96 (ours).

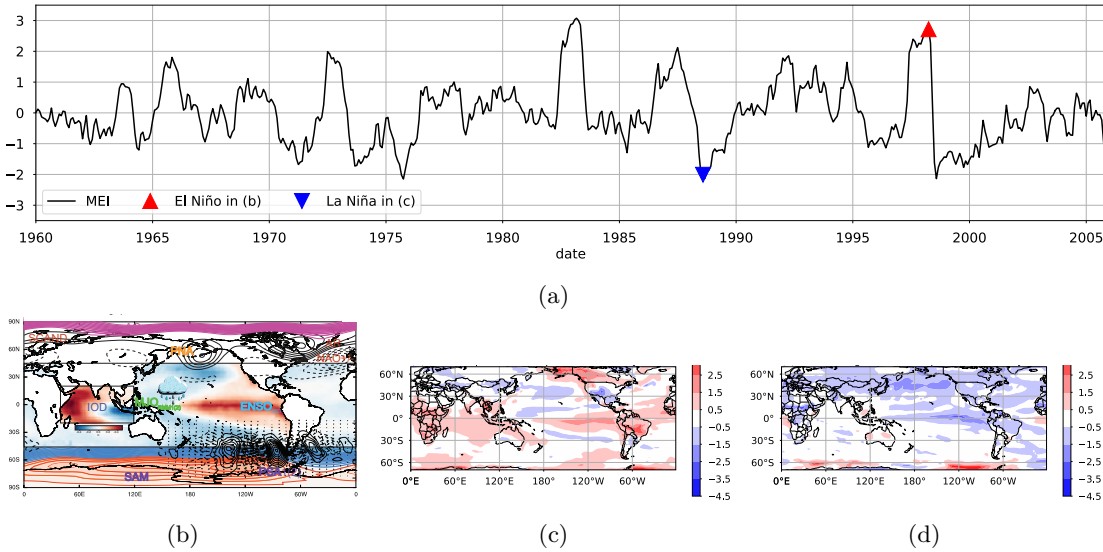

Figure 5: (a) MEI index. (b) The geographical locations of the indices. (c) El Niño temperature anomaly in April 1998. (d) La Niña temperature anomaly August 1988.

## E  More Analysis on Bayesian nonparametrics

To further analyze the shrinkage mechanism of our model with respect to the truncation level $K$, for each run (Figure 2), we obtain the matrix of factor weights $\{r_k^\tau\}_{k=1,\tau}^{K,\tau_{\max}}$, which we denote by $\boldsymbol{R} \in \mathbb{R}_+^{K,\tau_{\max}}$. For $K = 5$, we compute $r5 = \mathrm{median}(\boldsymbol{R})$. Since $K = 5$ is small, we use it as a reference point and define the number of effective latent factors $\ell_K$ as $\#\{r_k^\tau > r5\}_{k=1,\tau}^{K,\tau_{\max}}$.

Under this definition, we have $\ell_5 = \frac{5*\tau_{\max}}{2}$ by construction. For $K \in \{20, 50\}$, we report $\ell_K - \ell_5$ averaged over five runs with different random seeds. This quantity measures how much the number of effective latent

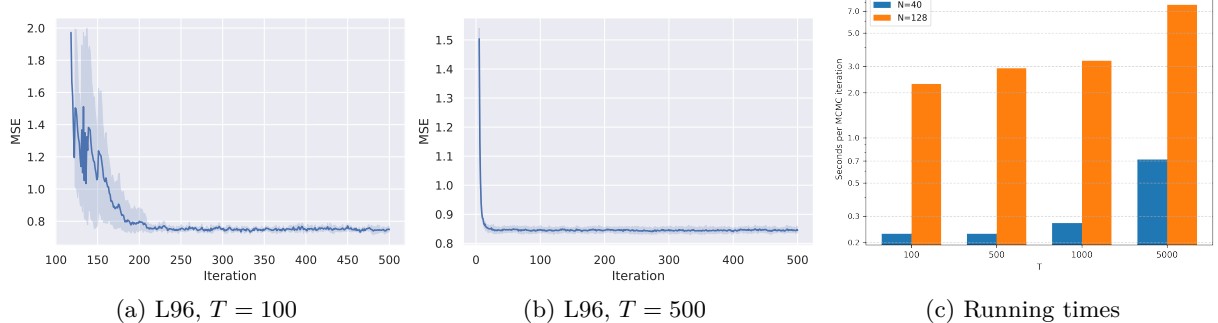

(a) L96, $T = 100$       (b) L96, $T = 500$       (c) Running times

Figure 6: (a, b) HiBaNG's ($V = 2$) empirical converge on Lorenz 96. For better visualisation, we show MSE in the iterations between $[a, b]$ where $a$ is the first iteration that MSE goes below 2.0 and $b = 500$. (c) Running times in seconds per MCMC iteration on on Lorenz 96.

Table 8: Difference in the number of effective latent factors, $\ell_K - \ell_5$, for $K \in \{20, 50\}$ across datasets and lag orders. The reference value $\ell_5$ is computed from the case $K = 5$.

| | L96, $T = 100$ | | L96, $T = 500$ | | LV, $T = 200$ | | LV, $T = 500$ | | FMRI | |
| | $K = 20$ | $K = 50$ | $K = 20$ | $K = 50$ | $K = 20$ | $K = 50$ | $K = 20$ | $K = 50$ | $K = 20$ | $K = 50$ |
|---|---|---|---|---|---|---|---|---|---|---|
| $\tau_{\max}=1$ | 0.000 | -0.667 | -0.667 | -0.333 | -0.333 | 0.000 | 1.667 | 4.000 | 0.000 | 0.000 |
| $\tau_{\max}=3$ | 0.667 | 2.000 | 2.667 | 2.000 | 2.333 | 0.667 | 2.000 | 2.667 | 0.000 | 2.667 |
| $\tau_{\max}=5$ | 2.000 | 3.000 | 1.333 | 4.667 | 3.000 | 1.000 | 2.000 | 2.667 | -1.333 | 0.667 |

factors deviates from the reference case $K = 5$, thereby indicating whether the model activates substantially more factors when the truncation level is increased.

We emphasize that $\ell_K$ is not intended to be a precise or exhaustive measure of the true number of active latent factors, but rather a coarse and interpretable proxy for assessing posterior shrinkage behavior. Since the notion of an "effective" factor is inherently ambiguous in hierarchical Bayesian models with continuous weights, any such metric is necessarily approximate. Our goal is therefore not exact quantification, but to provide a simple diagnostic that reflects whether increasing $K$ leads to a systematic increase in the number of activated components. As shown in Table 8, most values of $\ell_K - \ell_5$ remain small across all datasets, lag orders, and truncation levels. Even when increasing $K$ from 5 to 50, the number of effective latent factors grows only marginally and, in some cases, even decreases. This indicates that the posterior selectively activates only a limited number of latent factors, while the remaining capacity remains unused.

## F   Convergence Analysis

For a detailed convergence analysis of our algorithm, we set $V = 2$, $\tau_{\max} = 3$, and $K = 20$, and run 20 independent Gibbs sampling chains with different random seeds and initializations on the FMRI dataset. Consistent with our other experiments, each chain is run for 10,000 iterations, of which the first 5,000 are discarded as burn-in. From the remaining iterations, we retain one sample every 10 iterations, resulting in 500 samples per chain.

$\hat{R}$ **and ESS:** We first compute the posterior edge probabilities $\bar{p}$ (mean and standard deviation) for each pair of nodes at each lag by pooling the 500 post-burn-in samples across all chains. We then select a representative subset of 30 triplets $(i, j, \tau)$ (where $i$ and $j$ are node indices and $\tau$ is the lag index) with the largest posterior means for visualization. To assess convergence and mixing, we report the Gelman–Rubin diagnostic ($\hat{R}$) and effective sample size (ESS) for both $G$ and $A$ for the selected node pairs. Table 9 reports the convergence diagnostics. For both $G$ and $A$, $\hat{R}$ and ESS are close to 1, indicating reasonably good agreement across the 20 independent chains. Moreover, the effective sample sizes are consistently high and close to the total

number of retained samples, suggesting strong mixing and low autocorrelation. These results demonstrate that our Gibbs sampler achieves stable and reliable posterior inference for all selected interactions.

Table 9: Analysis of $\boldsymbol{G}$ and $\boldsymbol{A}$ in multiple chains on FMRI.

| $i$ | $j$ | $\tau$ | $\boldsymbol{G}$ | | | $\boldsymbol{A}$ | | |
|---|---|---|---|---|---|---|---|---|
| | | | $\bar{p}$ | $\hat{R}$ | ESS | $\bar{p}$ | $\hat{R}$ | ESS |
| 10 | 15 | 1 | $0.9001 \pm 0.29988$ | 1.0013 | 6380.5 | $-0.28935 \pm 0.70166$ | 0.99994 | 9000.7 |
| 9 | 8 | 2 | $0.3615 \pm 0.48046$ | 1.0002 | 7447.9 | $-0.11128 \pm 2.7951$ | 0.99959 | 9616.7 |
| 10 | 5 | 1 | $0.1468 \pm 0.35392$ | 1.0011 | 8421.7 | $-0.014161 \pm 2.5222$ | 1.0005 | 8795.9 |
| 12 | 11 | 1 | $0.118 \pm 0.32262$ | 1.0001 | 8672.4 | $0.043694 \pm 2.9617$ | 0.99935 | 9278.9 |
| 6 | 13 | 1 | $0.0924 \pm 0.2896$ | 1.0009 | 9453 | $0.012286 \pm 2.6168$ | 1 | 9663.2 |
| 2 | 10 | 1 | $0.0753 \pm 0.26389$ | 1.0001 | 9352 | $-0.018597 \pm 2.7518$ | 1 | 9383.4 |
| 14 | 15 | 1 | $0.074 \pm 0.26178$ | 0.99969 | 9083.2 | $0.029438 \pm 3.5335$ | 1.0003 | 9653.4 |
| 3 | 10 | 1 | $0.0725 \pm 0.25933$ | 0.9995 | 8889.1 | $-0.059261 \pm 2.7133$ | 0.9996 | 9507.9 |
| 1 | 11 | 1 | $0.0576 \pm 0.233$ | 1.0011 | 9375.3 | $-0.0049786 \pm 2.8795$ | 0.99953 | 9554.4 |
| 13 | 15 | 1 | $0.0548 \pm 0.2276$ | 1.0009 | 8844.8 | $0.070477 \pm 2.8283$ | 0.99949 | 9184.5 |
| 1 | 8 | 1 | $0.0529 \pm 0.22385$ | 1.0002 | 9159.9 | $0.026641 \pm 3.1418$ | 0.99943 | 9614 |
| 6 | 7 | 1 | $0.0515 \pm 0.22103$ | 0.99983 | 9802.8 | $-0.024542 \pm 4.5388$ | 1.0002 | 9473.8 |
| 13 | 6 | 1 | $0.0504 \pm 0.21878$ | 0.99947 | 9634.5 | $-0.043816 \pm 3.4608$ | 0.99991 | 9584 |
| 11 | 1 | 1 | $0.0462 \pm 0.20993$ | 0.99996 | 9065.2 | $-0.022 \pm 2.7709$ | 0.99933 | 9097.7 |
| 6 | 14 | 1 | $0.0424 \pm 0.20151$ | 0.99949 | 9337 | $0.019762 \pm 2.6693$ | 0.99984 | 9238.7 |
| 7 | 3 | 1 | $0.042 \pm 0.2006$ | 0.99935 | 9489.2 | $0.039643 \pm 3.1857$ | 0.99974 | 8766.6 |
| 2 | 4 | 1 | $0.041 \pm 0.1983$ | 0.99992 | 8932.8 | $-0.015108 \pm 4.9719$ | 0.99973 | 9161.6 |
| 10 | 4 | 1 | $0.0366 \pm 0.18779$ | 1.0003 | 9267 | $0.047841 \pm 3.3277$ | 0.99952 | 9003.4 |
| 10 | 8 | 1 | $0.0351 \pm 0.18404$ | 0.99998 | 9416.1 | $0.013438 \pm 2.571$ | 0.99941 | 9730.2 |
| 2 | 11 | 1 | $0.0338 \pm 0.18072$ | 1.0005 | 9180.9 | $0.057287 \pm 3.4617$ | 0.99993 | 9076.5 |
| 3 | 14 | 1 | $0.0323 \pm 0.1768$ | 1 | 9177.5 | $0.029325 \pm 3.4226$ | 0.99954 | 9608.5 |
| 7 | 14 | 2 | $0.0294 \pm 0.16893$ | 0.99944 | 9598.2 | $0.019568 \pm 4.3586$ | 1.0002 | 8831.7 |
| 8 | 10 | 1 | $0.027 \pm 0.16209$ | 1.0006 | 9494.6 | $-0.02755 \pm 2.5024$ | 0.99996 | 9472 |
| 8 | 6 | 1 | $0.0266 \pm 0.16092$ | 0.99939 | 9360.1 | $-0.012545 \pm 3.1591$ | 0.99953 | 9658.2 |
| 10 | 12 | 1 | $0.0249 \pm 0.15583$ | 1.0004 | 9446.7 | $0.033767 \pm 3.3515$ | 1.0004 | 9664.7 |
| 7 | 1 | 1 | $0.0245 \pm 0.1546$ | 0.99925 | 9573.5 | $-0.10686 \pm 11.404$ | 0.99977 | 9853.3 |
| 11 | 13 | 1 | $0.0234 \pm 0.15118$ | 1.0006 | 9555.4 | $0.016013 \pm 3.8499$ | 1.001 | 9381.4 |
| 5 | 4 | 2 | $0.0233 \pm 0.15086$ | 0.99982 | 9487.6 | $-0.034036 \pm 3.6502$ | 0.99944 | 8998.4 |
| 15 | 1 | 1 | $0.0224 \pm 0.14799$ | 1.0004 | 9047.4 | $-0.038182 \pm 4.2819$ | 0.99974 | 9340.3 |
| 14 | 13 | 1 | $0.0223 \pm 0.14766$ | 1.0012 | 9922.7 | $-0.044632 \pm 3.1499$ | 0.99968 | 9062.1 |

**Reliability diagram:** We evaluate the probabilistic calibration of posterior edge probabilities using reliability diagrams with respect to the ground-truth graph on the FMRI dataset. Due to the extreme sparsity of true edges, we construct bins with approximately equal numbers of edges to obtain stable empirical estimates. For each bin, we plot the mean predicted probability against the corresponding empirical edge frequency. Each reliability diagram is accompanied by a histogram of predicted probabilities to contextualize the calibration

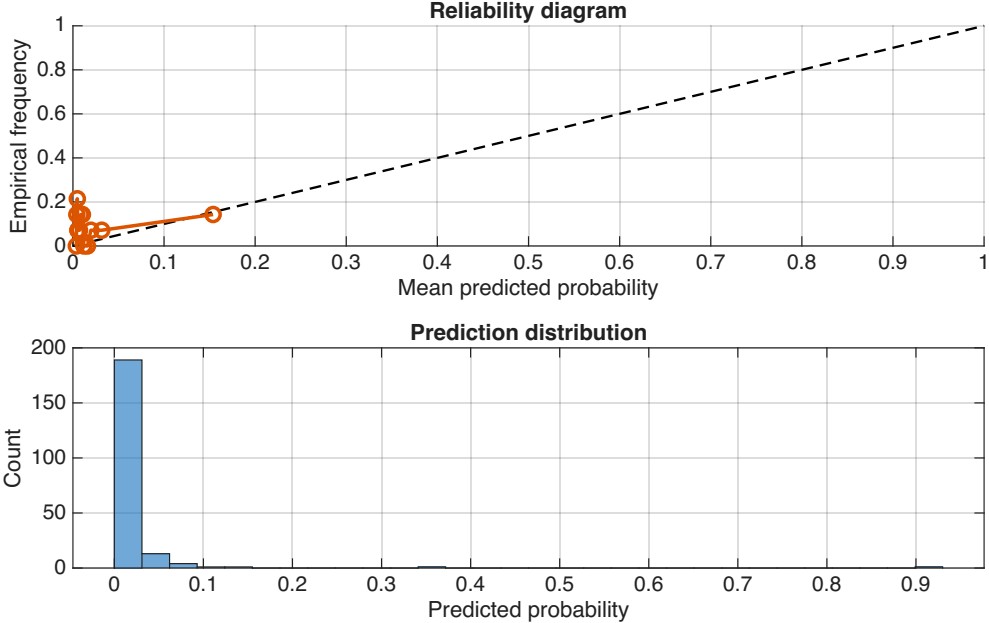

Figure 7: Reliability diagrams.

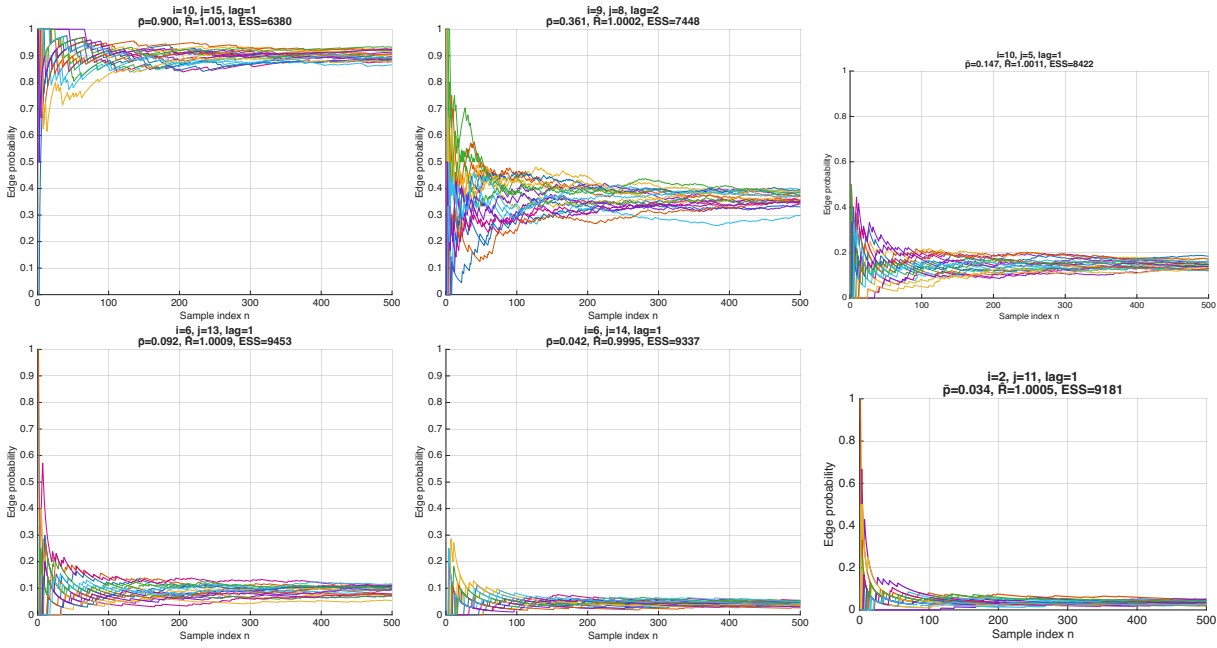

Figure 8: Edge probability traces. Colors indicate different chains.

curve. As shown in Figure 7, most predictions concentrate near zero, reflecting the sparsity of the underlying graph, while the calibration curve closely follows the diagonal in the low-probability regime, indicating reasonable calibration where most of the probability mass lies. This aligns with the CE results reported in Table 2.

**Edge probability traces:** To visually assess convergence and mixing of the Gibbs sampler, we include multi-chain trace plots for selected edge-specific variables. Because the latent variables are matrix-valued, it is not feasible to display traces for all entries. Instead, we visualize traces for a representative subset of

triplets $(i, j, \tau)$, selected previously. In Figure 8, for each selected triplet $(i, j, \tau)$, we plot its edge probability $\bar{p}$ across the collected 500 samples for all chains on the same axis. These trace plots provide a qualitative diagnostic of convergence, illustrating that all chains explore the same stationary distribution, exhibit no visible nonstationarity, and mix adequately. We complement these visual checks with quantitative convergence diagnostics shown in Table 9.

