# OpenReview forum: "HiBaNG: Hierarchical Bayesian Nonparametric Granger Causal Discovery in Low-Data Regimes"
_TMLR — Accepted by TMLR_

### Review · Reviewer_X8WU · 2025-11-14

**Summary Of Contributions:**

The paper introduces HiBaNG, a hierarchical Bayesian VAR for Granger‑causal discovery in low‑data settings. Coefficients are split into a binary lagged graph ($G_\tau$) and real‑valued weights ($A_\tau$). A new Generalized Bernoulli–Poisson Link (GBPL) thresholds a Poisson draw at level (V) to control graph sparsity; this enables a gamma–Poisson factorization prior over $G_\tau$. Inference uses a conjugate Gibbs sampler with per‑iteration cost $O(N^2)$ and total $O(N^2(V{+}K)\tau_{\max}+T\tau_{\max}^2)$. On several simulation and real-world tests using climate data, HiBaNG improves AUROC/AUPRC in most cases and shows lower calibration error on most of the settings; a climate (JRA‑55) case study illustrates interpretability.

Strengths
- Design is reasonable and reproducible: principled uncertainty over edges, simple and transparent inference, good performance in scarce‑data regimes
- Clear write‑up with hyperparameter grids and clear intuition derivation for the proposed algorithm.

Weaknesses
- Model selection uses test‑set MSE
- sensitivity to key hyperparameters is thin
- noise covariance is diagonal (not realistic)
- real‑data validation beyond the climate case is limited (i.e. lacks broader real-world impact validations)

**Audience:**

Yes

**Audience Explanation:**

The paper tackles a common pain point "causal discovery with short time series" using a compact Bayesian model that returns calibrated edge probabilities. The GBPL‑based prior and the separation of $G_\tau$ from $A_\tau$ are likely to interest both causal‑discovery and time‑series researchers.

**Broader Impact Concerns:**

The paper has a Broader Impact section.
Two items need clearer emphasis:
- Granger != interventional causality; results hinge on stationarity and unconfounded lags, so edges should be vetted by domain experts, especially in high‑stakes domains.
- For human or sensitive data, note bias/privacy risks and basic mitigations.

**Claims And Evidence:**

Yes

**Claims Explanation:**

- Methods and assumptions are explicit with sufficient references to past works.
- The Gibbs updates and complexity are derived clearly.
- Across benchmarks, HiBaNG is typically best or second‑best in AUROC/AUPRC; on FMRI it ranks second (PCMCI+ is first), thus indicating strong support for the statements made in the paper.
- The experiment setups are well elaborated (e.g. graphical illustrations and tabular results in the appendix, results are computed in multiple runs with mean/variance calculated).

**Requested Changes:**

- If possible, please replace test‑set tuning with a validation split or rolling origin CV for all methods
- Report R‑hat/ESS or multi‑chain traces for key variables and include reliability diagrams for edge probabilities (beyond CE)
- Vary $V$, $K$, and $\tau_{\max}$ for additional ablation studies. You can also discuss the effect of the diagonal covariance assumption (e.g. compare with full covariance)
- Fix an inconsistency: the text/Appendix tune $V\in \{1,2,3\}$, but Table 3 shows $V=\{1,3,5\}$.

---

> ### Author Response · Authors · 2026-01-16
>
> We thank the reviewer for the insightful comments, which are addressed by either our responses as follows or in the revisions. We highlighted the revised parts in our submission in blue.
>
> ## Weaknesses
>
> 1. Please see our response to Requested Change 1
>
> 2. Please see our response to Requested Change 3
>
> 3. We acknowledge that assuming a diagonal noise covariance is a simplification and may not fully capture all forms of contemporaneous dependence in real-world systems. Our motivation for this choice is twofold. First, it isolates temporal Granger-causal effects from instantaneous correlations, which is standard practice in Granger causality modeling. Second, it substantially simplifies inference and improves identifiability, especially in low-data regimes, where jointly estimating a full noise covariance and a dynamic causal structure is often ill-posed. We do not claim that this assumption is universally realistic. Rather, it is a deliberate modeling choice that trades off expressiveness for interpretability, robustness, and computational tractability.
>
> 4. We appreciate the reviewer’s point regarding broader real-world validation. We respectfully note that fully real-world datasets rarely provide ground-truth causal graphs, making quantitative evaluation inherently difficult and often requiring domain-expert interpretation. Our choice of climate reanalysis data was deliberate: this domain offers well-established, physically grounded causal relationships (e.g., ENSO teleconnections), enabling us to perform a qualitatively meaningful assessment of causal plausibility. Moreover, we conducted extensive synthetic and semi-synthetic experiments to emulate realistic dynamics while providing ground-truth causal structure, allowing rigorous quantitative evaluation.
>
> ## Requested Changes:
>
> 1. We thank the reviewer for this suggestion. In standard supervised learning settings, splitting data into train/validation/test sets is indeed common practice. However, our problem is fundamentally unsupervised: we aim to discover Granger-causal graph structures without access to ground-truth graphs during training. This setup is more analogous to unsupervised tasks such as clustering or topic modeling, where model selection cannot rely on labeled validation data. Because our goal is to infer global structural properties rather than optimize pointwise prediction accuracy, there is no natural notion of a “test set” in the conventional supervised sense. Instead, we use forecasting performance (e.g., MSE on held-out temporal segments) as a proxy objective for model selection. This choice is consistent with the definition of Granger causality itself, which is inherently based on predictive improvement. Importantly, this procedure is applied consistently across all VAR-based baselines and our method, ensuring a fair comparison. We have revised "Model Selection and Parameter Tuning" to make it clearer.
>
> 2. We have added new results and analysis to comprehensively exam the convergence of our algorithm in Section E of the appendix, including the suggested metrics.
>
> 3. We have added additional ablation study in the appendix to address the comment.
>
> 4. Thanks for the comment. We have fixed the issue in the revision.

---

### Review · Reviewer_Y2nF · 2025-12-05

**Summary Of Contributions:**

The paper addresses the problem of discovering Granger causality among variable in multivariate times-series using Bayesian nonparametric framework. Focusing on the low-data regime, they propose a sparsity-encouraging prior for the underlying causal graphs and develops an Gibbs sampling algorithms for inference.
The proposed method is evaluated and  compared to several baseline for synthetic and real data, but there is not theoretical results.

The paper makes the following main claims:
1) The proposed prior is claimed to enable "interpretable, uncertainty aware inference"
2) The resulting algorithm is claimed to improve accuracy and calibration
3) The resulting algorithm reduces overfitting
4) The resulting algorithm has a reduced number of tunable hyperparameters.

Generally, the paper is well-written, well-structured, and easy to follow. However, section 3 and 5 could be significantly improved, see comments below.

**Additional Comments:**

The intro says: "The difficulty of model selection without access to sufficient validation data or ground-truth causal graphs, making hyperparameter tuning unreliable and potentially biased and, ultimately, compromising robustness and interpretability". Since this is stated as a fact, please support it by citations or own evidence/arguments.

The intro says: "Introducing a hierarchical factorized prior over binary Granger causal graphs, which encodes structured sparsity and incorporates inductive bias in the absence of abundant data". All models/methods have inductive biases, the question is whether they are useful or not. It would be great with a more detail description of the induced inductive bias.

The Bayesian baseline BVAR is from 1986. Is this really the state of the art for Bayesian approach to Granger causality?

**Audience:**

Yes

**Audience Explanation:**

Causality is an important topic and I am sure the a subset of the TMLR audience would be interested in work like this.

**Claims And Evidence:**

No

**Claims Explanation:**

Ideally I would have answered: "Yes to no claims" and "no for other" above. I'll elaborate in the following:

Regarding claim 1)
The abstract, introduction, and conclusion highlight interpretability in various ways, but it is completely ignored in the rest of the paper. Therefore, this claim is not supported. Moreover, it is somewhat unclear what "interpretable, uncertainty-aware inference" even means.

Regarding claim 2)
This claim is supported by the results in Table 1 & 2, showing improved accuracy as measured by AUROC and calibration. However, it is unclear to me how strong the baselines are.

Regarding claim 3)
The method is claimed to reduce overfitting, and the method does indeed perform well compared to the baseline, but it is unclear whether this is caused by reduced overfitting or something else. Using "great performance on train set, poor perfomance on val/test set" as a loosse definition for overfitting, I don't see much evidence for this claim.

Regarding claim 4)
It is claimed that the method has a reduced number of tunable hyperparameters. However, the proposed prior has several hyperparameters, e.g. K, c, d_k, e_k. The paper claims that the results are not sensitive to K, but there is no evidence to support this. Moreover, it is also unclear how sensitive the results are to the remaining hyperparameters.

**Requested Changes:**

Overall, the work seems solid, but there are several things in the presentation and the results that makes it harder to appreciate. I'll elaborate on some of the things that would improve the paper in my opinion below.

Notation, symbols, abbreviations are in many places not defined explicitly, e.g.
- In eq. (1), does A^\tau refer to the matrix A raised to the power of \tau or does \tau index a collection of matrices?
- Property 3.3 mentions a "conditional posterior of m", but it is unclear what it is supposed to be conditioned on.
- Which parametrization of the gamma distribution is used in eq. (3)?
- The metric SHD appears only to be defined in the appendix.

The presentation of the method in sec. 3 appears unclear and unmotivated in several places. E.g.
-  "we first present our BVAR model in a general form, which separates the coefficients into binary GC graphs and weight matrices".
- "we propose a new link function named generalized Bernoulli Poisson link (GBPL) that thresholds a random Poisson variable m at V ∈ {1, 2, . . . } to obtain a binary variable b, as one of the key building blocks in our approach."
- Why is it useful to separate graph connections from strength? What is the authors' definition of the a link function? Do they refer to a link function from generalized linear models or something else? What spaces are the link function defined? What problem does it solve?  I can guess some of the answer, but the paper would be improved if the authors would explain, motivate, and justify the construction.

Section 3.4 highlights that the model has fewer hyperparameters, and that a user in essence only has to tune V. The prior has several hyperparameters, e.g. K, c, d_k, e_k and V. How sensitive is the resulting method to use? How does the performance of the method behave as a function of K?

The proposed prior resembles an Indian Buffet process (IBP) in any ways. It would be great if the authors could compare their proposed prior to the IBP.

The abstract, introduction, and conclusion highlight interpretability, but it is largely ignored in the rest of the paper. The author should provide some insight into this aspect.

**Experiments**

- Section 5 simply states that the authors focus on AUROC for evaluation, but one needs to go the appendix to figure out what the AUROC is computed for (e.g. link in the graph, where links are estimated using the posterior mean of the Bernoulli variables). Such information needs to be in the main paper in my opinion.
- Also it is unclear to me what the MSE is measuring. Please elaborate on this.
- While the authors do show MSE as a function of iterations, it would be useful to know whether the chains have mixed (e.g. via the potential scale reduction factor statistic or similar) and  what the effective samples sizes are etc.
- It is unclear what the "uncertainties" in Table 1&2 represents. It is standard error, standard deviation across the 5 runs or something else?
- How is the ground truth graph defined for the FMRI dataset?

---

> ### Author Response · Authors · 2026-01-16
>
> We thank the reviewer for the insightful comments, which are addressed by either our responses as follows or in the revisions. We highlighted the revised parts in our submission in blue.
>
> ## Claims
>
> 1: Interpretability and “uncertainty-aware inference”
>
> We agree that the notion of interpretability and uncertainty-aware inference should be made more explicit and operational in the paper. By interpretable, we mean that the model infers explicit, sparse Granger causal graphs whose edges have a clear semantic meaning: a directed edge from variable $i$ to $j$ at lag $\tau$ corresponds to a direct, lag-specific causal influence in the underlying VAR dynamics. This contrasts with many Deep VAR approaches, where causal structure is often implicit in neural network weights and difficult to inspect or summarize.
>
> By uncertainty-aware inference, we mean that our method infers a posterior distribution over graph structures rather than a single point estimate. This allows us to quantify epistemic uncertainty over edges which is not typically available in deterministic Deep VAR approaches.
>
> 2. Baselines
>
> We appreciate the reviewer’s concern regarding the strength of the baselines. Our experimental evaluation includes both classical Bayesian VAR models and state-of-the-art neural VAR–based Granger causal discovery methods. This design choice was because: classical Bayesian VARs provide well-established probabilistic baselines that are commonly used in low-data regimes, while neural VAR models represent recent high-capacity approaches that emphasize expressive nonlinear dynamics. By comparing against methods from both categories, we aim to provide a comprehensive evaluation across different modeling paradigms.
>
> 3. Overfitting
>
> We thank the reviewer for this clarification. In this work, we use the term overfitting in a general and informal sense, consistent with common usage in the machine learning literature: namely, that models with a large number of free parameters and weak regularization tend to fit spurious patterns and noise in low-data regimes, leading to poorer generalization. Since our method addresses an unsupervised causal discovery problem, there is no standard notion of training–test loss curves. In our experiments, we observe that when the amount of available data decreases, competing methods tend to exhibit a more pronounced degradation in performance, whereas our method remains relatively stable. We use this behavior as empirical evidence of reduced overfitting in low-data regimes.
>
> 4. Hyperparameters
>
> We would like to clarify the roles of the hyperparameters in our model. The parameter $K$ serves as a truncation level for the underlying Bayesian nonparametric construction, rather than as a conventional model complexity parameter. In truncated nonparametric models, $K$ is usually chosen sufficiently large so that the posterior mass concentrates on a much smaller effective number of active components, making the results largely insensitive to its exact value.
>
> We would like to clarify that the quantities $a_i$, $b_j$, $d_k$, $e_k$ are not fixed hyperparameters in our model, but rather random variables that are part of the Bayesian hierarchy and are inferred from the data. These parameters appear at the top levels of the hierarchy and govern global properties such as scale and sparsity. Following standard practice in hierarchical Bayesian and Bayesian nonparametric models, we place weakly informative (Gamma(1,1)) priors on these variables in order to avoid imposing strong structural assumptions and to allow the appropriate degree of regularization to be learned adaptively from the data. As a result, these quantities do not act as performance-critical tuning knobs; instead, they enable automatic regularization and reduce sensitivity to manual parameter choices.

---

> > ### Author Response · Authors · 2026-01-16
> >
> > ## Requested Changes:
> >
> > 1. Notation, symbols, abbreviations
> >
> > Thanks for the comments. We have revised the paper accordingly (highlighted in blue of the revision)
> >
> > 2. "The presentation of the method in sec. 3 appears unclear and unmotivated in several places"
> >
> > In Section 3, we actually first gave a complete picture of our proposed model and then introduced the motivations of our design choices at the end of the section. We believe Section 3.4 is able to address the concerns on clarity and motivations. We have revised the first paragraph of Section 3 to clarify the flow of the presentation of the section.
> >
> > 3. Hyperparameters
> >
> > Please see our responses to the comment in "Claims".
> >
> > 4. Indian Buffet process (IBP)
> >
> > Thank you for pointing out the connection to the Indian Buffet Process (IBP). We agree that IBP is related to our approach in that both define Bayesian nonparametric priors over binary matrices. However, the two models are developed for different purposes. Specifically, IBP is typically used to model a binary asymmetric feature-assignment matrix (e.g., indicating which latent features are active for each object), whereas our goal is to define a structured prior over a binary symmetric adjacency matrix representing a Granger causal graph.
> >
> > Our model is built on the framework of Poisson factorization, which is primarily motivated by considerations of inference and scalability. In particular, this construction enables conditionally conjugate updates and efficient Gibbs sampling when embedded within the full time-series model. While it may be possible in principle to adapt IBP-based constructions to our setting, to the best of our knowledge, this direction remains underexplored, likely due to the additional modeling and inference constraints that arise when coupling IBP-style priors with dynamical systems and graphs.
> >
> > 5. Interpretability
> >
> > Please see our responses to the comment in "Claims".
> >
> > 6. Experiments
> > - We have moved the details of the experimental settings to the main paper in Section 5.1 as suggested.
> > - We have added detailed explanation of MSE in "Model selection and parameter tuning" in Section 5.1.
> > - We have added new results and analysis to comprehensively exam the convergence of our algorithm in Section E of the appendix.
> > - It is standard deviation across the 5 runs. We have added a sentence to clarify this in Section 5.1.
> > - To address the comment, we have clarified how the ground-truth graph is generated for the FMRI dataset in Section 5.2 of the revision.
> >
> > ## Additional comments
> >
> > 1. We have revised the introduction and added references to support our statement, as suggested.
> > 2. We have revised the introduction to make it clearer.
> > 3. Our primary goal is to compare against recent neural VAR–based Granger causality methods, which are often proposed as modern alternatives to classical VAR-based approaches. The inclusion of classical Bayesian VARs is not meant to suggest that they are state of the art, but rather to provide a well-understood Bayesian reference point. This allows us to isolate the effect of explicitly modeling binary causal graphs and structured sparsity within a Bayesian framework.

---

> > ### Comment · Reviewer_Y2nF · 2026-01-16
> > **Thank you for the clarification**
> >
> > Thank you for the clarifications.
> >
> > I am happy with the clarifications in 1) and 2).
> >
> > I think my point regarding 3) still stands since your summary of overfitting is consistent with my definition and if you cannot quantify the degree of overfitting, it is hard to make claims about it. But we agree that you observe better generalization.
> >
> > About 4), I am fully aware that the K is the truncation level and that posteriors of BNP-models are often can be encourage to concentrate on a small and relevant subsets, especially for "nice and well-behaved" problems. My point here is that you claim the method is insensitive to K, but you are not providing into evidence for that claim for your specific set-up.
> >
> > And about the hyperparameters $a_i$, $b_j$, $d_k$, $e_k$ -  sorry, I completely missed the line: "noninformative gamma priors Gamma (1, 1) are used for ai, bj , dk, ek, and c.".
> > Thank you for the clarification.

---

> > > ### Author Response · Authors · 2026-01-19
> > >
> > > We sincerely thank the reviewer for their thoughtful follow-up and for their positive assessment of our responses to points (1) and (2).
> > >
> > > Regarding (3), given the unsupervised nature of Granger causal discovery, we agree with the reviewer that it is inherently difficult to precisely quantify overfitting. While we report predictive performance and uncertainty calibration in the paper, these should indeed be interpreted as indirect indicators rather than direct measurements of overfitting. To improve rigor and avoid overclaiming, we have revised the manuscript to reduce the use of the term “overfitting” and to more carefully qualify related statements.
> > >
> > > Regarding (4), as expected for a stochastic Bayesian model with multiple interacting latent components, predictive performance exhibits some variability as $K$ changes. However, our empirical results show that this variability remains within a narrow and stable range, with no systematic trend or degradation as $K$ increases. To further examine the role of
> > > $K$, we have added a new analysis in Section E of the appendix, where we report a proxy measure of the number of active latent factors for different truncation levels. This provides additional empirical evidence supporting the truncation-based interpretation of $K$.
> > >
> > > All new revisions are highlighted in red in the updated manuscript.

---

### Review · Reviewer_gDsH · 2025-12-17

**Summary Of Contributions:**

This paper introduces **HiBaNG**, a hierarchical Bayesian nonparametric framework for Granger causal discovery in multivariate time series, with a particular focus on low-data regimes where deep learning–based VAR models often struggle. The key contribution is an explicit probabilistic separation between binary Granger causal graphs and real-valued VAR coefficients, enabled through a Poisson-factorized hierarchical prior and a generalized Bernoulli–Poisson link function. This design allows structured sparsity, uncertainty-aware inference, and tractable Gibbs sampling. The methodology is carefully developed and theoretically motivated, and the experimental evaluation, especially on synthetic and semi-synthetic benchmarks, is thorough and convincing. At the same time, the approach relies on a stylized linear VAR assumption, which might not be useful in real-world applications. It introduces modeling choices (e.g., lag-wise independent graph priors and hierarchical Poisson constructions) whose necessity and generality would benefit from further clarification and justification.

**Additional Comments:**

This is a well-written paper with clearly stated assumptions and thoughtful justifications supporting its main claims. The statistical methodology is carefully presented, and the experimental evaluation, particularly on synthetic and semi-synthetic datasets, is comprehensive and well executed. That said, the paper has several conceptual and modeling limitations, particularly regarding temporal assumptions, linearity, and the necessity of hierarchical graph priors, which are detailed in the requested changes above.

**Audience:**

Yes

**Audience Explanation:**

Yes, this paper is of high significance to the causal discovery, VAR, Granger causality, and time-series community. Granger causality is also tied to root cause analysis/diagnosis for systems with time-series data. Therefore, researchers working on anomaly detection would also benefit from this analysis. However, the restriction of the models to linear statistical methods with closed-form might limit their applicability to only controlled synthetic datasets. The authors need to work on the extension of this work to Deep methods for supporting real-world industrial applications.

**Broader Impact Concerns:**

No concerns about the ethical implications of this work. The authors have done a good job mentioned a detailed "Broader Impact Statement".

**Claims And Evidence:**

Yes

**Claims Explanation:**

The main claim of the paper, namely that the proposed hierarchical Bayesian nonparametric framework enables effective Granger causal discovery in low-data regimes, is supported by convincing empirical evidence. The authors present a comprehensive experimental evaluation on synthetic, semi-synthetic, and real-world datasets. In particular, the synthetic and semi-synthetic experiments are well controlled and directly test the stated advantages of the method in low-sample settings. That said, while the empirical results are strong, several modeling choices and claims would be further strengthened by more meticulous theoretical analysis and justification, particularly regarding temporal assumptions and prior construction.

**Requested Changes:**

**(1) Independence of Granger graphs across time lags**

In the multiple-lag setting, the paper states that “we have a separate generative process for the GC graph at each lag $\tau$.” This appears to imply that the prior factorizes as
$$
p(G^1, \dots, G^{\tau_{\max}}) = \prod_{\tau=1}^{\tau_{\max}} p(G^\tau),
$$
i.e., that the Granger causal graphs at different lags are *a priori* independent. Please clarify whether this interpretation is correct. If so, the authors should provide a clearer mathematical justification for why independent priors across time lags are reasonable, especially given that many real-world dynamical systems exhibit consistent or decaying causal effects across multiple delays. A short theoretical argument or discussion explaining why likelihood-based posterior coupling alone is sufficient (or preferable) would strengthen the modeling rationale.

**(2) Linear VAR assumption vs. Deep VAR models**

The proposed framework is built on a linear Gaussian VAR likelihood, which is a stylized assumption that is rarely satisfied in complex real-world systems. In contrast, much of the recent state of the art in Granger causal discovery relies on nonlinear or neural-network-based VAR models (Deep VAR). While the paper provides an extensive and thoughtful literature review of Deep VAR methods, it remains unclear whether HiBaNG should be viewed primarily as (i) an interpretable simplification of Deep VAR approaches, or (ii) a fundamentally different modeling paradigm with distinct theoretical advantages.

At a minimum, the paper would benefit from a dedicated discussion or section outlining how the proposed hierarchical Bayesian graph prior could be extended to nonlinear or neural VAR settings, for example by replacing the linear predictor
$$
x_t = \sum_{\tau} (A^\tau \odot G^\tau)x_{t-\tau}
$$
with a parameterized nonlinear function while retaining the same probabilistic treatment of $G^\tau$. This section should include an appropriate mathematical formulation. Additionally, 1–2 sentences should be added to the Deep VAR portion of the related work to explicitly articulate how HiBaNG differs in modeling assumptions and inference goals.

Relatedly, please clarify whether any preprocessing or transformations were applied to the real-world climate data to justify a linear VAR assumption. If no such preprocessing was used, does the paper implicitly claim that a linear VAR provides an adequate fit to the raw climate indices? This point requires clarification.

**(3) Justification for the hierarchical Bayesian graph prior**

The motivation for adopting a hierarchical Bayesian construction over Granger causal graphs remains insufficiently justified. It is unclear what would fail, theoretically or empirically, if a simpler prior were used, such as
$$
G^\tau_{i,j} \sim \mathrm{Bernoulli}(\pi), \quad \text{or} \quad \pi \sim \mathrm{Beta}(a,b).
$$
The authors should provide theoretical insight or formal arguments explaining why such Bernoulli (or Beta–Bernoulli) graph models are inadequate in this setting, and why the proposed Poisson-factorized hierarchical construction is necessary. Without this justification, the methodology risks appearing as a collection of sophisticated statistical components rather than a principled modeling choice grounded in theory.

---

> ### Author Response · Authors · 2026-01-16
>
> We thank the reviewer for the insightful comments, which are addressed by either our responses as follows or in the revisions. We highlighted the revised parts in our submission in blue.
>
> 1. Independence of Granger graphs across time lags
>
> To address the comment, we have added a dedicated discussion in Section 3.4 of the revision.
>
>
> 2. Linear VAR assumption vs. Deep VAR models
>
> - Our use of a linear Gaussian VAR likelihood is a deliberate modeling choice rather than an oversight. Instead of being a drop-in replacement for Deep VAR approaches, our method is better to be viewed as a complementary paradigm that emphasizes interpretability, uncertainty quantification, and robustness in low-data regimes. In contrast to neural VAR models, which typically require large sample sizes and produce point estimates of causal structure, our framework provides a fully Bayesian posterior directly over binary Granger graphs with uncertainty.
> - We have added a dedicated discussion in Section 3.4 of the revision outlining how our model can be potentially extended to nonlinear settings.
> - We have added sentences in Section 4 (Related Work) to better articulate how our model compares with Deep VARs in terms of assumptions and goals.
> - For the climate reanalysis data, we note that many of the climate indices derived from reanalysis products are themselves low-dimensional summaries of high-dimensional spatiotemporal physical fields (e.g., geopotential height, sea-level pressure, wind, and outgoing longwave radiation). Although some steps in their construction are linear, these indices represent compressed, hand-crafted representations of complex and inherently nonlinear climate dynamics. While we do not claim that a linear VAR provides a fully faithful generative model of the underlying climate system, we believe that adopting a linear assumption is reasonable in this context. Our goal is not to maximize predictive expressiveness, but rather to provide a qualitative case study that highlights the interpretability and uncertainty-aware causal discovery capabilities of our framework.
>
> 3. Justification for the hierarchical Bayesian graph prior
>
> We agree that it is possible to do $G^\tau_{i,j} \sim \text{Bernoulli}(\pi)$ where $\pi \sim \text{beta}(a, b)$ is a natural choice. However, this might not fit our setting:
>
> - Hierarchical modelling: Our aim is to build a factorized, hierarchical, and nonparametric Bayesian model, where dependencies across time lags and variable pairs can be shared and structured through latent factors. This kind of expressiveness is difficult to achieve with a simple Beta-Bernoulli prior while preserving local conjugacy.
> - Inference efficiency: Inference in such hierarchical Bernoulli models typically requires more complex algorithms (e.g., rejection sampling, Metropolis-Hastings, or variational inference).
>
> To address these issues, we build the Bernoulli distribution by truncating a Poisson distribution at $\pi$, which leverages the well-established theory of Poisson matrix factorisation with several benefits:
> - It allows us to model sparsity directly via $\pi$, while also enabling analytically tractable inference within a hierarchical model.
> - The Poisson construction enables factorisation of the graph structure in an elegant way, allowing us to introduce shared latent variables that capture common patterns across time lags and variables.

---

> ### Comment · Reviewer_gDsH · 2026-01-18
> **Response to Authors' Rebuttal**
>
> Thank you for the detailed rebuttal.
>
> (1) The concern raised in point (1) is well clarified in the revised manuscript.
>
> (2) The extension to nonlinearity is explained clearly and is reasonable. However, the following statement in the rebuttal raises some concerns: *“While we do not claim that a linear VAR provides a fully faithful generative model of the underlying climate system, we believe that adopting a linear assumption is reasonable in this context.”* In particular:
>
> (a) How can one assess whether the learned model provides an accurate representation of the true (Granger) causal structure? Fitting a linear VAR to data generated by nonlinear dynamics may introduce biases or misspecifications, and the possible consequences of this mismatch should be discussed more explicitly.
>
> (b) Is the baseline PCMCI+ also based on linear construction steps? If not, one would expect the proposed model to be disadvantaged in comparison. In this light, I was somewhat surprised by the claim: *“These are consistent with Harries & O’Kane (2021) (Figures 6 and 7), and not captured by PCMCI. Overall, the HiBaNG graph shows a larger and more diverse set of edges consistent with those previously inferred from the JRA55 reanalysis via Bayesian structure learning.”* While it is certainly possible to obtain empirically stronger results, the argument would benefit from additional justification grounded in the underlying mathematics or theory to explain why such improvements should be expected.
>
> (3) I am not fully convinced by the justification for hierarchical modeling over a simpler prior. In particular, could you clarify the statement from the rebuttal: *“Our aim is to build a factorized, hierarchical, and nonparametric Bayesian model, where dependencies across time lags and variable pairs can be shared and structured through latent factors.”* Why is this a central aim of the paper? From my perspective, the primary objective is to develop a principled Bayesian approach to learning Granger causality. I am unsure how the emphasis on shared latent structure directly supports this goal, and some clarification would help.
>
> **That said, I find the paper to be strong and impressive overall. My comments are intended to ensure that the key claims are carefully justified and articulated, so that the contribution is compelling for the broader community.**

---

> ### Author Response · Authors · 2026-01-19
>
> We sincerely thank the reviewer for their positive assessment of our work and for recognizing its overall strength. We greatly appreciate the careful and constructive nature of the comments, which have helped us refine our claims, improve clarity, and better articulate the scope and limitations of our contributions. We address each point below.
>
> Please let us address each point in the additional replies in turn.
>
> ### For (2)
>
> (a) We agree that the sentence in the rebuttal was too strong. A linear VAR is not a faithful generative model for many real-world systems, including the climate system, and we did not intend to imply otherwise. We also acknowledge that fitting a linear model to nonlinear data may introduce bias. Our motivation for using a linear formulation in potentially nonlinear settings is twofold:
>
> - When the true system is nonlinear, linear Granger-based methods may still recover meaningful aspects of the causal structure if dependencies are approximately linear in expectation, or if nonlinear effects induce detectable linear predictive improvements.
>
> - Our primary target setting is the low-data regime, a linear approximation can be preferable, as it may potentially yield more stable posterior inference and better-calibrated uncertainty than highly flexible nonlinear models.
>
> We have revised the manuscript to refine our claims in Section 3.4.
>
> b. PCMCI(+) is not restricted to linear models; it supports nonlinear conditional independence tests (e.g., kNN-based CMI, GP-based tests) and is therefore more expressive in principle than a linear VAR. We would like to clarify that the climate experiment is intended primarily as a qualitative demonstration of our framework’s interpretability and practical applicability, with PCMCI serving as a reference method. We did not intend to claim that HiBaNG should systematically outperform PCMCI on climate data. We have revised the manuscript to soften any language that could be interpreted as a claim of superiority.
>
> We have revised in Section 5.3 accordingly.
>
> ### For (3)
>
> By referring to shared latent structure in the rebuttal, we meant that our hierarchical construction enables information sharing across variables, which can stabilize inference through partial pooling and impose a structured inductive bias over graphs. This can be relevant in low-data regimes, where posterior inference is inevitably more strongly influenced by the prior. However, we agree with the reviewer that the primary objective of our work is to develop a principled Bayesian framework for Granger causal discovery, and not to emphasize the shared latent structure as an end in itself. We have checked the manuscript to ensure our claims are appropriately framed.
>
> All new revisions are highlighted in red in the updated manuscript.

---

### Decision · Action_Editor_DkXy · 2026-01-30

**Recommendation:** Accept as is

**Audience:**

Yes

**Audience Explanation:**

Yes, causality is an active domain of interest in TMLR and this paper brings a new perspective and new results.

**Claims And Evidence:**

Yes

**Claims Explanation:**

The claims made by the paper are backed by convincing empirical evidence in both synthetic and semi-synthetic domains.